# ViCo: Plug-and-play Visual Condition for Personalized Text-to-image Generation

## Abstract

Personalized text-to-image generation using diffusion models has recently emerged and garnered significant interest. This task learns a novel concept (*e.g.*, a unique toy), illustrated in a handful of images, into a generative model that captures fine visual details and generates photorealistic images based on textual embeddings. In this paper, we present ViCo, a novel lightweight plug-and-play method that seamlessly integrates visual condition into personalized text-to-image generation. ViCo stands out for its unique feature of not requiring any fine-tuning of the original diffusion model parameters, thereby facilitating more flexible and scalable model deployment. This key advantage distinguishes ViCo from most existing models that necessitate partial or full diffusion fine-tuning. ViCo incorporates an image attention module that conditions the diffusion process on patch-wise visual semantics, and an attention-based object mask that comes at no extra cost from the attention module. Despite only requiring light parameter training ($\sim 6\%$ compared to the diffusion U-Net), ViCo delivers performance that is on par with, or even surpasses, all state-of-the-art models, both qualitatively and quantitatively. This underscores the efficacy of ViCo, making it a highly promising solution for personalized text-to-image generation without the need for diffusion model fine-tuning.

## 1 Introduction

Nowadays, people can easily generate unprecedentedly high-quality photorealistic images with text prompts using fast-growing text-to-image diffusion models (Ho et al., 2020; Song et al., 2021; Ramesh et al., 2022; Nichol et al., 2022; Saharia et al., 2022; Rombach et al., 2022). However, these models are trained on a text corpus of seen words, and they fail to synthesize novel concepts like a special-looking dog or your Batman toy collection. Imagine how fascinating it would be if your plastic Batman toy could appear in scenes of the original 'Batman' movie. Recent works (Gal et al., 2023a; Ruiz et al., 2023; Kumari et al., 2023) make this fantasy come true, terming the task *personalized* text-to-image generation. Specifically, given several images of a unique object, the goal is to capture the object and reconstruct it in text-guided image generation.

DreamBooth (Ruiz et al., 2023) incorporates a unique identifier before the category word in the text embedding space and finetunes the entire diffusion model during training. The authors also finetune the text encoder, which empirically shows improved performance. Custom Diffusion (Kumari et al., 2023) finds that only tuning a few parameters, *i.e.*, key and value projection matrices, is sufficiently powerful. DreamBooth and Custom Diffusion both meet the issue of language drift (Lee et al., 2019; Lu et al., 2020) because finetuning the pretrained model on new data can lead to a loss of the preformed language knowledge. They leverage a preservation loss to address this problem, which requires manually generating (Ruiz et al., 2023) or retrieving massive class-specific images. Textual Inversion (Gal et al., 2023a) adopts minimal optimization by exclusively learning a novel text embedding to represent the given object, showing enhanced performance using latent diffusion models (Rombach et al., 2022). For the more powerful Stable Diffusion, however, the learned embedding struggles to express fine details of the visual object, and the generated results are prone to overfitting to training samples due to the limited fine-grained expressiveness of CLIP (Radford et al., 2021). In this work, we follow (Gal et al., 2023a) to use a single learnable token embedding $S_\star$ to represent the novel concept instead of the form of "[V] class" used in (Ruiz et al., 2023; Kumari et al., 2023). In our vision, a single token embedding should be capable of effectively representing any visual concept within an ideal unified text-image space.

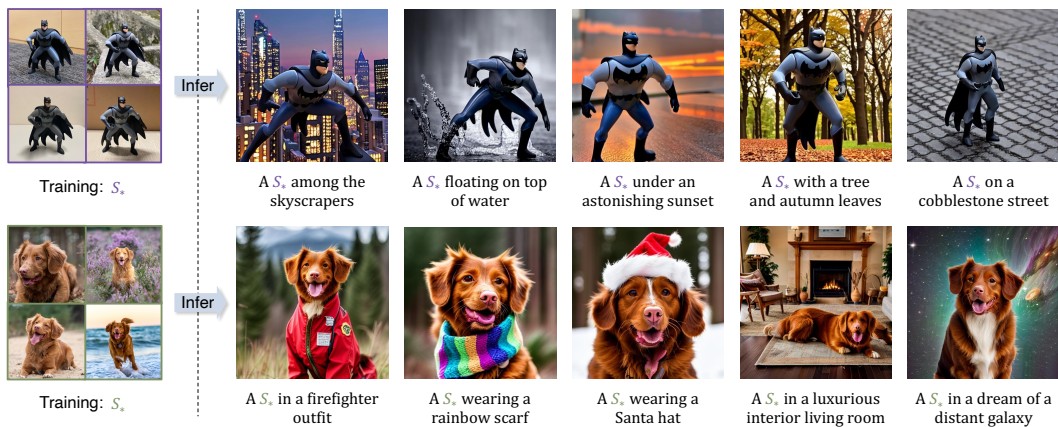

Figure 1: **Personalized text-to-image generation.** Generated images of the Batman toy (top) and the Toller (bottom) by `ViCo`. $S_\star$ denotes the learnable text embedding (Gal et al., 2023a).

To overcome the issue of declined model expressiveness of novel concepts, we propose a novel plug-in method that integrates visual conditions into the diffusion process, which harnesses the inherent richness of diffusion models. Specifically, we present an image cross-attention module, facilitating the seamless integration of intermediate features from a reference image, which are extracted by the original denoising U-Net, into the denoising process. Our method stands out for not requiring any modifications or fine-tuning of any layers in the original diffusion model, setting it apart from most existing methods like DreamBooth (Ruiz et al., 2023) and Custom Diffusion (Kumari et al., 2023). As the language knowledge remains intact without requiring fine-tuning of the diffusion model, our method avoids the problem of language drift, which eliminates the need for heavy preprocessing like image generation (Ruiz et al., 2023) and retrieval (Kumari et al., 2023).

Another challenge we address is the difficulty in isolating the foreground object of interest from the background. Instead of relying on prior annotated masks as in concurrent works (Wei et al., 2023; Shi et al., 2023; Jia et al., 2023), we propose an automatic mechanism to generate object masks that are naturally incorporated into the denoising process. Specifically, we leverage the notable semantic correlations between text and image in cross-attentions (Hertz et al., 2023) and utilize the cross-attention map associated with the learnable object token to generate an object mask. Our method is computationally efficient, non-parametric, and online, and can effectively suppress the influence of distracting backgrounds in the training samples. We also design an easy-to-employ regularization between the cross-attention maps associated with the end-of-text token and the learnable token to help refine the object mask.

We name our model `ViCo`, which offers a number of advantages over previous works. (1) It is fast (∼6 minutes) and lightweight (6% of diffusion U-Net). (2) It is plug-and-play and requires no fine-tuning of the original diffusion model, allowing highly flexible and transferable deployment. (3) It is easy to implement and use, requiring no heavy preprocessing or mask annotations. (4) It can preserve fine object-specific details of the novel concept in text-guided generation (see Fig. 1).

Our contributions include: (1) proposing an image cross-attention module to integrate visual conditions into the denoising process for capturing object-specific semantics; (2) introducing an automatic object mask generation mechanism from the cross-attention map; (3) providing quantitative and qualitative comparisons with state-of-the-art methods (Ruiz et al., 2023; Kumari et al., 2023; Gal et al., 2023a) and demonstrating the efficiency of `ViCo` in multiple applications.

## 2 RELATED WORK

**Text-to-image synthesis.** In the literature of GANs (Goodfellow et al., 2014; Brock et al., 2019; Karras et al., 2019; 2020; 2021), plenty of works have gained remarkable progress in text-to-image generation (Reed et al., 2016; Zhu et al., 2019; Tao et al., 2022; Xu et al., 2018; Zhang et al., 2021; Ye et al., 2021) and image manipulation using text (Gal et al., 2022; Patashnik et al., 2021; Xia et al., 2021; Abdal et al., 2022), advancing the generation of images conditioned on plain text. These methods are trained on a fixed dataset that leverages strong prior knowledge of a specific domain. Towards a zero-shot fashion, auto-regressive models (Ramesh et al., 2021; Yu et al., 2022) trained on large-scale data of text-image pairs achieve high-quality and content-rich text-to-image generation results. Based on the pretrained CLIP (Radford et al., 2021), Crowson *et al.* (Crowson

et al., 2022) applies CLIP similarity to optimize the generated image at test time without any training. The use of diffusion-based methods (Ho et al., 2020) has pushed the boundaries of text-to-image generation to a new level. Examples include DALL·E 2 (Ramesh et al., 2022), Imagen (Saharia et al., 2022), GLIDE (Nichol et al., 2022), and LDM (Rombach et al., 2022). Recently, some works consider personalized text-to-image generation by learning a token embedding (Gal et al., 2023a) and finetuning (Ruiz et al., 2023) or partially finetuning (Kumari et al., 2023) a diffusion model. Recently, Qiu et al. (2023) proposes a fine-tuning method named Orthogonal Finetuning, which can be used for efficient DreamBooth fine-tuning. Many works emerge lately, but they require finetuning the whole or partial networks in the vanilla U-Net such as Perfusion (Tewel et al., 2023), ELITE (Wei et al., 2023), and UMM-Diffusion (Ma et al., 2023), or training with large-scale data on specific category domains based on encoders (Shi et al., 2023; Jia et al., 2023; Gal et al., 2023b; Chen et al., 2023) or with text-image pairs (Li et al., 2023). In contrast, our work tackles the general domain-agnostic task while keeping the pretrained diffusion model completely frozen. We compare the characteristics of different models in Tab. 1.

Table 1: **Model characteristics.**

|  | Placeholder type | Preprocessing | #Trainable params | Diffusion U-Net | Text encoder | Visual condition |
|---|---|---|---|---|---|---|
| DreamBooth (Ruiz et al., 2023) | [V] class | Generation | 982.6M | Fully finetuned | Finetuned | ✗ |
| Custom Diffusion (Kumari et al., 2023) | [V] class | Retrieval | 57.1M | Partially finetuned | Frozen | ✗ |
| Textual Inversion (Gal et al., 2023a) | $S_\star$ | Null | 768 | Frozen | Frozen | ✗ |
| ViCo | $S_\star$ | Null | 51.3M | Frozen | Frozen | ✓ |

**Visual condition.** Visual condition is commonly used in image-to-image translation (Isola et al., 2017; Zhu et al., 2017a;b; Choi et al., 2018; Park et al., 2020), which involves training a model to map an input image to an output image based on a certain condition, *e.g.*, edge, sketch, or semantic segmentation. Similar techniques have been used for tasks such as style transfer (Gatys et al., 2016; Johnson et al., 2016), colorization (Zhang et al., 2016; Larsson et al., 2016; Zhang et al., 2017), and super-resolution (Ledig et al., 2017; Johnson et al., 2016; Wang et al., 2018). In the context of diffusion models, visual condition is also used for image editing (Brooks et al., 2023) and controllable conditioning (Mou et al., 2023; Zhang & Agrawala, 2023). Despite the massive study on visual condition, most works use it for controlling the spatial layout and geometric structure but discard its rich semantics. Our work stands out in capturing fine-grained semantics related to the specific visual appearance from visual conditions, an aspect that is rarely discussed.

**Diffusion-based generative models.** Diffusion-based generative models develop fast and continuously produce striking outcomes. Ho *et al.* (Ho et al., 2020) first presents DDPMs to progressively denoise from random noise to a synthesized image. DDIMs (Song et al., 2021) accelerate the sampling process of DDPMs. Latent diffusion models (LDMS) (Rombach et al., 2022) introduce multiple conditions in latent diffusion space, producing realistic and high-fidelity text-to-image synthesis results. Following the implementation of LDMs (Rombach et al., 2022), Stable Diffusion (SD) is trained on a large-scale text-image data collection, which achieves the state-of-the-art text-to-image synthesis performance. Diffusion models are widely used for generation tasks such as video generation (Ho et al., 2022; Wu et al., 2022), inpainting (Lugmayr et al., 2022), and semantic segmentation (Hoogeboom et al., 2021; Baranchuk et al., 2022).

## 3 METHOD

Given a handful of images (4-7) showing a novel object concept, we target at generating images of this unique object following some text guidance. We aim to neatly inject visual condition, which is neglected in previous works, along with text condition into the diffusion model to better preserve the visual expressions. Following the attempt of textual inversion (Gal et al., 2023a), we adopt a placeholder ($S_\star$) as the learnable text embedding to capture the unique visual object. We first quickly review Stable Diffusion (Rombach et al., 2022) which serves as our base model (Sec. 3.1). We then introduce a simple yet efficient method to inject fine-grained semantics from visual conditions into the denoising process (Sec. 3.2), and show how to automatically generate object masks within training (Sec. 3.3). We finally present our overall learning objective and implementation details (Sec. 3.4). Fig. 2 shows an overview of our method.

### 3.1 STABLE DIFFUSION

Stable Diffusion (SD) (Rombach et al., 2022) is a latent text-to-image diffusion model derived from classic Denoising Diffusion Probabilistic Models (DDPMs) (Ho et al., 2020). SD applies a largely pretrained autoencoder $\mathcal{E}$ to extract latent code for images, and a corresponding decoder $\mathcal{D}$

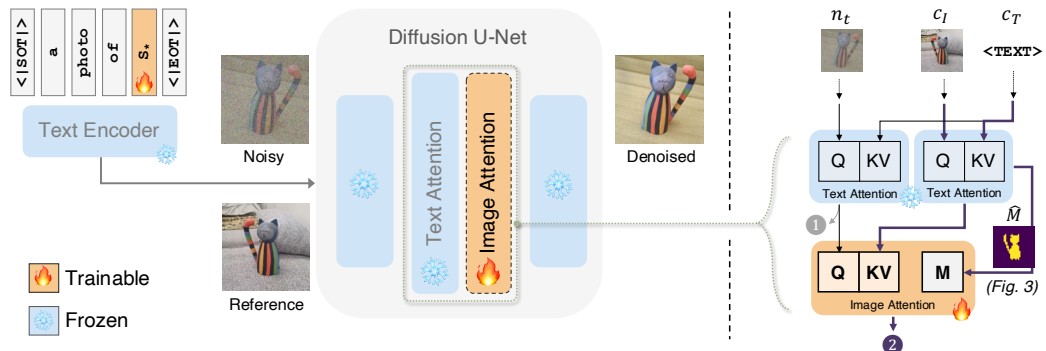

Figure 2: **Method overview.** We introduce a module of image (cross-)attention to integrate visual conditions into the frozen diffusion model. On the left, the noisy image and a reference image are fed into diffusion U-Net in parallel. We follow (Gal et al., 2023a) to learn the embedding $S_\star$. On the right, we present the data stream comprising the original text attention and the proposed image attention. ❶ denotes the attention output in vanilla diffusion model and ❷ represents the visually conditioned output. The generation and use of the mask $\hat{M}$ are further detailed in Sec. 3.3.

to reconstruct the original images. Specifically, the autoencoder maps images $x \in \mathcal{I}$ to latent code $z = \mathcal{E}(x)$, and the decoder maps latent code back to images $\hat{x} = \mathcal{D}(\mathcal{E}(x))$, where $\hat{x} \approx x$. SD adopts a diffusion model in the latent space of the autoencoder. For the text-to-image diffusion model, text conditions can be added to the diffusion process. The diffusion process can be formulated as iterative denoising that predicts the noise at the current timestep. In this process, we have the loss

$$\mathcal{L}_{SD} = \mathbb{E}_{z \sim \mathcal{E}(x), y, \epsilon \sim \mathcal{N}(0,1), t}[\|\epsilon - \epsilon_\theta(z_t, t, c_\pi(y))\|_2^2] \tag{1}$$

where $t$ is the timestep, $z_t$ is the latent code at timestep t, $c_\pi$ is the text encoder that maps text prompts $y$ into text embeddings, $\epsilon$ is the noise sampled from Gaussian distribution, and $\epsilon_\theta$ is the denoising network (*i.e.*, U-Net (Ronneberger et al., 2015)) that predicts the noise. Training SD is flexible, such that we can jointly learn $c_\pi$ and $\epsilon_\theta$ or exclusively learn $\epsilon_\theta$ with a frozen pretrained text encoder.

### 3.2 VISUAL CONDITION INJECTION

Common approaches for conditioning diffusion models on images include feature concatenation (Brooks et al., 2023) and direct element-wise addition (Mou et al., 2023; Zhang & Agrawala, 2023). These visual conditions show astonishing performance in capturing the layout of images. However, visual semantics, especially fine-grained details, are hard to preserve or even lost using these image conditioning methods. Instead of only considering the patches at the same spatial location on the noisy latent code and visual condition, we exploit correlations across all patches on both images. To this end, we propose to train an image cross-attention block that has the same structure as the text cross-attention block in the vanilla diffusion U-Net. The image cross-attention block takes an intermediate noisy latent code and a visual condition as inputs, integrating visual conditions into the denoising process.

Some works (Gal et al., 2023b; Shi et al., 2023) acquire visual conditions from reference images by additionally training a visual feature extractor. This may cause a misalignment between the feature spaces of the latent code and the visual condition. Instead of deploying extra networks, we directly feed the autoencoded reference image into the vanilla diffusion U-Net, and apply the intermediate latent codes as visual conditions. We use the pretrained autoencoder to map the reference image $x_r \in \mathcal{I}$ to the latent space: $z_r = \mathcal{E}(x_r)$. Let $\epsilon_\theta^l(\cdot)$ denote the output of the $l$-th attention block of U-Net. The visual condition at the $l$-th attention block is then given by

$$c_I^l = \epsilon_\theta^l(z_r, t, c_T), l \in \{0, 1, \cdots, L-1\} \tag{2}$$

where $L$ is the number of attention blocks in U-Net, and $c_T = c_\pi(y)$ is the text condition from the text encoder. Note that $c_T$ is derived from token embeddings in which the embedding $S_\star$ is learnable. Let the raw text cross-attention block from vanilla U-Net be $\mathcal{A}_T(q, kv)$, the proposed image cross-attention block be $\mathcal{A}_I(q, kv)$. We denote the new denoising process after incorporating $\mathcal{A}_I$ as $\epsilon_{\theta,\psi}$, in which the $l$-th attention block is denoted as $\epsilon_{\theta,\psi}^l$. We can compute the intermediate latent code of the generated noisy image at the $l$-th attention block as

$$n_t^l = \epsilon_{\theta,\psi}^l(z_t, t, c_T, c_I^l), l \in \{0, 1, \cdots, L-1\} \tag{3}$$

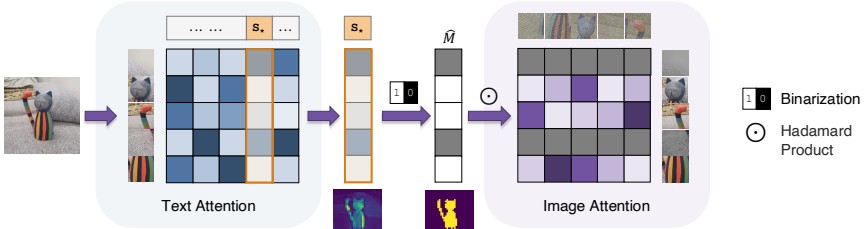

Figure 3: **Mask mechanism.** We can obtain a similarity distribution from the cross-attention map of the reference image associated with the learnable object token $S_\star$. The distribution can be unflattened into a similarity map. After binarization with Otsu thresholding (Otsu, 1979), the derived binary mask can be applied to the image cross-attention map to discard the non-object patches.

Because all operations are executed at the $l$-th attention block, we can omit all superscripts of $l$ for simplicity. The original attention at each attention block in U-Net, *i.e.*, $n_t' = \mathcal{A}_T(n_t, c_T)$, can be replaced by $\hat{n}_t = \mathcal{A}_T(n_t, c_T)$, $\hat{c}_I = \mathcal{A}_T(c_I, c_T)$, $n_t' = \mathcal{A}_I(\hat{n}_t, \hat{c}_I)$, where $n_t'$ is the output of the current attention block that is fed into the following layers in U-Net. At the image cross-attention block $\mathcal{A}_I$, we can capture visual semantics from the reference image and inject them into the noisy generated image.

### 3.3 EMERGING OBJECT MASKS

To avoid capturing the background from training samples and exclusively learn the foreground object we are interested in, we propose an online, computationally-efficient, and non-parametric method that is naturally incorporated into our pipeline to generate reliable object masks. Next, we will illustrate how attention maps of text and image conditions can be directly used as object masks to capture the object-exclusive patch regions.

Recall the process of computing the text cross-attention in diffusion U-Net

$$\texttt{TextAttention}(Q, K, V) = \texttt{softmax}(\frac{QK^T}{\sqrt{d_k}})V \tag{4}$$

where the query is the reference image, the key and value are the text condition, and the scaling factor $d_k$ is the dimension of the query and key.[1] Inspired by (Hertz et al., 2023) that diffusion models gain pretty good cross-attentions, we notice the attention map $\texttt{softmax}(\frac{QK^T}{\sqrt{d_k}})$ inherently implies a good object mask for the reference image. Specifically, the attention map for the text condition and the visual condition from the reference image reveals the response distribution of each text token to all image patches of different resolutions. The learnable embedding $S_\star$ has strong responses at the exact patch regions where the foreground object lies. After binarizing the similarity distribution of $S_\star$ on the reference image, we can obtain a good-quality object mask $\hat{M}$. In this paper, we simply apply Otsu thresholding (Otsu, 1979) for binarization. The mask can be directly deployed in our proposed image cross-attention by simply masking the attention map between the noisy generated image and the reference image in the latent space. The masked image cross-attention is formulated as

$$\texttt{ImageAttention}(Q, K, V) = \left( \hat{M} \odot \texttt{softmax}(\frac{QK^T}{\sqrt{d_k}}) \right) V \tag{5}$$

where the query is the noisy generated image, the key and value are the reference image, and $\odot$ is Hadamard product (element-wise product) with broadcasting $\hat{M}$. The masking process is depicted in Fig. 3. By masking the attention map in Eq. (5), distractors from the background can be drastically suppressed. We can thus condition the generation process exclusively on the foreground object that is captured in the reference image.

**Regularization.** Due to fine-tuning on a small set of images, $S_\star$ is sometimes overfitted, deriving undesirable object masks. Nevertheless, we empirically find the end-of-text token `<|EOT|>`, the global representation in transformers, can maintain consistently good semantics on the unique object. From this observation, we apply a regularization between similarity maps of the reference image associated with $S_\star$ and `<|EOT|>` in the text cross-attention. Specifically, from cross-attentions, we have the attention map $A := \texttt{softmax}(\frac{QK^T}{\sqrt{d_k}}) \in \mathbb{R}^{B \times D_p \times D_t}$ where $B$ is the batch size, $D_p$ is

---

[1] All happen in the latent space after linear projections.

the number of image patches, and $D_t$ is the number of text tokens. Let $S_\star$ be the $i$-th token and `<|EOT|>` be the $j$-th token, and their corresponding similarity logits be $A_{\star,i}$ and $A_{\star,j}$. We define our regularization as

$$\mathcal{L}_{reg} = \|A_{\star,i}/\max(A_{\star,i}) - A_{\star,j}/\max(A_{\star,j})\|_2^2 \tag{6}$$

where we apply a max normalization to guarantee the same scale of the two logits. We can flexibly leverage this regularization during training to refine the attention map of $S_\star$, thereby enhancing the object mask used in our method. This refinement ensures the reliability of the mask.

### 3.4 Training and inference

**Training.** We train our model on 4-7 images with the vanilla diffusion U-Net frozen. We formulate the final training loss by integrating the standard denoising loss and the regularization term as

$$\mathcal{L} = \mathbb{E}_{z\sim\mathcal{E}(x),z_r\sim\mathcal{E}(x_r),y,\epsilon\sim\mathcal{N}(0,1),t}[\|\epsilon - \epsilon_{\theta,\psi}(z_t,t,z_r,c_\pi(y))\|_2^2] + \lambda\mathcal{L}_{reg} \tag{7}$$

where $\lambda$ is the scaling weight of the regularization loss, and $\epsilon_{\theta,\psi}$ is the new denoising networks composed of the vanilla diffusion U-Net parameterized by $\theta$ and the proposed image attention blocks parameterized by $\psi$. During training, we freeze the pretrained diffusion model and only train the image attention blocks and finetune the learnable text embedding $S_\star$ simultaneously.

**Implementation details.** We use Stable Diffusion (Rombach et al., 2022) as our backbone. The diffusion U-Net (Ronneberger et al., 2015) contains encoder, middle, and decoder layers. We incorporate the proposed image attention module into every other attention block exclusively in the decoder. Our image attention module follows the standard attention-feedforward fashion (Vaswani et al., 2017), and has the same structure as the text cross-attention used in LDMs (Rombach et al., 2022) only differing in the dimension of the condition projection layer. We set $\lambda = 5 \times 10^{-4}$ and learning rate to $5 \times 10^{-3}$ for $S_\star$ and $10^{-5}$ for image attention blocks. We train `ViCo` with a batch size of 4 for 400 steps. At inference, our model also requires a reference image input for the visual condition, injected into the denoising process in the same way as in training. Our method is insensitive and robust to the reference image. Therefore, either one in the training samples or a new image of the identical object is a feasible visual condition in sampling. For fair evaluation, we use the same reference image for each dataset in all experiments.

## 4 Experiment

### 4.1 Quantitative evaluation

**Data.** Previous works (*e.g.*, Textual Inversion (Gal et al., 2023a), DreamBooth (Ruiz et al., 2023), and Custom Diffusion (Kumari et al., 2023)) use different datasets for evaluation. For a fair and unbiased comparison, we collect a dataset of 20 unique concepts from these three works. The collected dataset spans a large range of object categories covering 6 toys, 6 live animals, 4 accessories, 3 containers, and 1 building, allowing a comprehensive evaluation. Each object category contains 4-7 images of a unique object (except for one having 12 images). Based on the prompt list provided in (Ruiz et al., 2023), we remove one undesirable prompt "a cube shaped $S_\star$" because we are more interested in keeping the appearance of the unique object. In addition, we add more detailed and informative prompts to test the expressiveness of richer and more complex textual knowledge (*e.g.*, "a $S_\star$ among the skyscrapers in New York city"). Totally, we collect 31 prompts for 14 non-live objects and 31 prompts for 6 live animals. We generate 8 samples per prompt for each object, giving rise to 4,960 images in total, for robust evaluation. More details about the dataset can be found in Appendix B.

**Metric.** In our task, we concern with two core questions regarding the personalized generative models: (1) how well do the generated images capture and preserve the input object? and (2) how well do the generated images tail the text condition? For the first question, we adopt two metrics, namely CLIP (Radford et al., 2021) image similarity $I_{\text{CLIP}}$ and DINO (Caron et al., 2021) image similarity $I_{\text{DINO}}$. Specifically, we compute the feature similarity between the generated image and the corresponding real image respectively using CLIP (Radford et al., 2021) or DINO (Caron et al., 2021). DINO is trained in a self-supervised fashion without ground-truth class labels, thus not neglecting the difference among objects from the same category. Therefore, DINO metric better reflects how well the generated object resembles the real one, as also noted in (Ruiz et al., 2023). For the second question, we adopt one metric, namely CLIP text similarity $T_{\text{CLIP}}$. Specifically, we compute the feature similarity between the CLIP visual feature of the generated image and the CLIP textual feature of the corresponding prompt text that omits the placeholder. The three metrics are derived from the average similarities of all compared pairs. In our experiments, we deploy ViT-B/32 for the CLIP vision model and ViT-S/16 for the DINO model to extract visual and textual features.

Table 2: **Quantitative comparison.**

| Quantitative metrics | $I_{\text{DINO}}\uparrow$ | $I_{\text{CLIP}}\uparrow$ | $T_{\text{CLIP}}\uparrow$ |
|---|---|---|---|
| DreamBooth (Ruiz et al., 2023) | 0.628 | 0.804 | 0.236 |
| Custom Diffusion (Kumari et al., 2023) | 0.570 | 0.768 | **0.249** |
| Textual Inversion (Gal et al., 2023a) | 0.520 | 0.768 | 0.216 |
| ViCo | **0.631** | **0.809** | 0.229 |

Table 3: **Time cost (averaged over 5 runs).**

| Time cost (sec.) | Training | Inference |
|---|---|---|
| DreamBooth (Ruiz et al., 2023) | 1411±27 | 11.2±0.1 |
| Custom Diffusion (Kumari et al., 2023) | 682±59 | **8.4**±0.4 |
| Textual Inversion (Gal et al., 2023a) | 735±7 | 9.8±0.4 |
| ViCo | **353**±3 | 15.4±0.1 |

**Comparison.** We compare our method `ViCo` with three state-of-the-art models, namely Textual Inversion (Gal et al., 2023a), DreamBooth (Ruiz et al., 2023), and Custom Diffusion (Kumari et al., 2023). We use Stable Diffusion for all compared methods for a fair comparison. The results of three quantitative metrics are shown in Tab. 2. Our model achieves the highest image similarity on both DINO and CLIP metrics, indicating our method best preserves the object-specific semantics from the image. DreamBooth and Custom Diffusion perform better on the text similarity metric because they use the fashion of "[V] class" to represent the visual object in the text space. The class category word provides rich prior knowledge, while the learnable identifier "[V]" primarily serves as an auxiliary guidance, such as controlling texture or facial appearance in the generation process. In contrast, Textual Inversion and our method employ a single token in the text embedding space, which, once learned, may dominate the text space and slightly weaken the influence of text-related information in the generated results. We deliberately choose the single-token fashion in our work because we believe that representing a visual concept with a single word token is crucial for achieving effective text-image alignment. This minimalist approach allows us to capture the essence of the concept in a concise and precise manner, focusing on the core problem of aligning textual and visual information. Besides, DreamBooth and Custom Diffusion require finetuning either the full SD network or a portion of it while our model and Textual Inversion do not. With the same foundation, our method outperforms Textual Inversion by a significant margin on all metrics.

We report the training and inference time cost of the four methods in Tab. 3. All methods are trained using four 3090 GPUs and tested on a single one. Note that DreamBooth requires generating coarse-category samples and Custom Diffusion involves retrieving real images with given and similar captions, which are not included in the time overheads presented in the table. Overall, the majority of the time cost is in the training, while the inference takes much less time for all methods. `ViCo` has a slightly longer inference time due to the additional image attention.

### 4.2 QUALITATIVE EVALUATION

In our massive qualitative experiments, depicted in Fig. 4, we observe that `ViCo` produces text-guided images of high quality. We assess the qualitative results based on several aspects.

**Image fidelity.** Our model preserves fine details of the object in the training samples. As a comparison, Textual Inversion fails to preserve sufficient details in many cases (the 3rd and 5th rows) due to its limited expressiveness. The use of "[V] class" in DreamBooth and Custom Diffusion, while providing strong class-related information, may result in the loss of object-specific details. For instance, in the second row, both DreamBooth and Custom Diffusion alter the appearance of the cat. Similarly, DreamBooth fails to preserve the holes in the generated elephant in the fourth row.

**Text fidelity.** Our model can faithfully follow the text prompt guidance to generate reasonable results. For example, in the first row of the "teddy bear", our model successfully incorporates elements such as "a tree" and "autumn leaves" as indicated by the text, while other models may occasionally struggle to achieve this level of fidelity. In more complex cases, like the third and fourth rows, Textual Inversion fails to express any information from the text prompts.

**Text-image Equilibrium.** Our model excels at balancing the effects of both text conditions and visual conditions, resulting in a harmonious equilibrium between the text and the image. The text prompts and the image samples may have varying degrees of influence on generation. For example, in the last row, Custom Diffusion successfully generates an appealing "lion face" guided by the text, but the generated image is almost no longer a "pot". Similarly, DreamBooth maintains the overall appearance of a pot but loses significant details of the original "wooden pot". In contrast, our method excels at preserving the original "pot" details while synthesizing a high-quality "lion face" on it.

**Authenticity.** Our generation results are authentic and photorealistic, devoid of noticeable traces of artificial synthesis. For example, in the fourth row, although Custom Diffusion generates visually appealing images, they may appear noticeably synthetic. In comparison, our results are more photorealistic, authentically depicting a golden elephant statue positioned at Times Square.

**Diversity.** Our model demonstrates the capability to generate diverse results, presenting a notable abundance of variation and showcasing a wide range of synthesis possibilities.

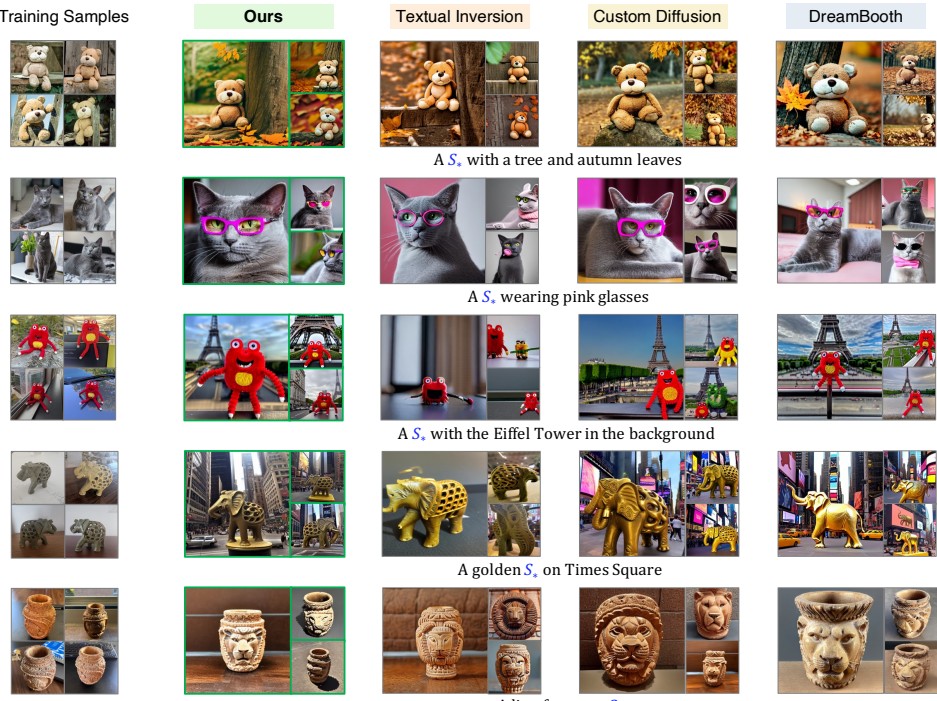

Figure 4: **Qualitative comparison.** Given input images (first column), we generate three samples using ViCo (ours), Textual Inversion (Gal et al., 2023a), Custom Diffusion (CD) (Kumari et al., 2023), and DreamBooth (DB) (Ruiz et al., 2023). The text prompt is under the generation samples, in which $S_\star$ for CD and DB is "[V] class".

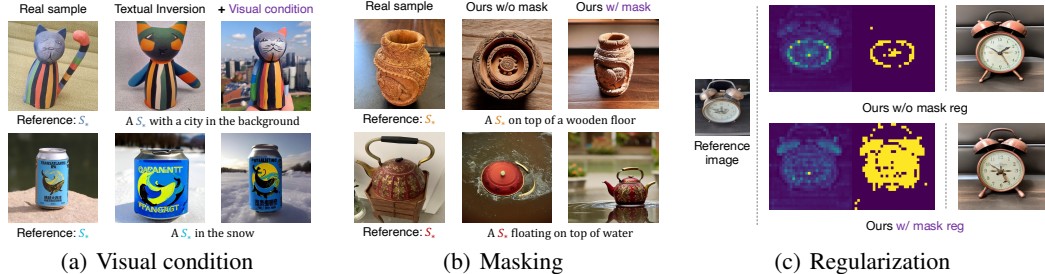

|  (a) Visual condition | (b) Masking | (c) Regularization |
|---|---|---|

Figure 5: **Ablation study.** We ablate each component in our method and report: (a) results with or without the visual condition; (b) results with or without the masking; and (c) attentions, masks, and generations (from left to right) with or without the regularization.

## 4.3   ABLATION STUDY AND ANALYSIS

We study the effect of the visual condition, the automatic mask, and the initialization of $S_\star$. Representative results are compiled in Fig. 5 and a quantitative comparison is reported in Tab. 4.

**Visual condition.**   The proposed visual condition module can significantly improve the visual expressiveness of the single learnable embedding used by Textual Inversion, making higher image fidelity. We compare the performance of Textual Inversion before and after adding the visual condition module in Fig. 5(a). We can observe the degree of object detail preservation is considerably enhanced without losing text information after adding our visual condition module. Row (1) in Tab. 4 also shows our visual condition can significantly enhance our baseline Textual Inversion (Gal et al., 2023a).

**Automatic mask.**   Our automatic mask mechanism enables isolating the object from the distracting background, which further improves the object fidelity. As shown in Fig. 5(b), the generation results may be occasionally distorted without the mask. After adding the mask, the object can be well captured and reconstructed. Row (2) in Tab. 4 also quantitatively shows applying the mask can further improve image fidelity. We also validate using regularization for the object mask refinement in Fig. 5(c), showing the mask is well aligned with the object after leveraging the regularization term.

Table 4: **Quantitative improvements.** TI denotes the baseline Textual Inversion (Gal et al., 2023a), VC denotes our visual condition, and M denotes the proposed mask mechanism.

| | TI | VC | M | $I_{\text{DINO}} \uparrow$ | $I_{\text{CLIP}} \uparrow$ | $T_{\text{CLIP}} \uparrow$ |
|---|---|---|---|---|---|---|
| (0) | ✓ | | | 0.520 | 0.768 | 0.216 |
| (1) | ✓ | ✓ | | 0.630 +21.2% | 0.805 +4.8% | **0.229** +6.0% |
| (2) | ✓ | ✓ | ✓ | **0.631** +21.3% | **0.809** +5.3% | **0.229** +6.0% |

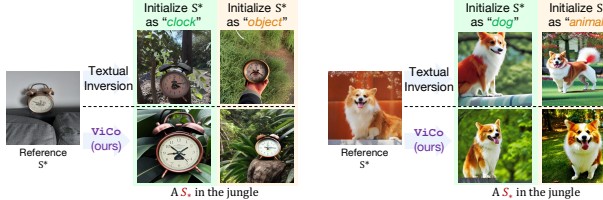

Figure 6: Comparison of our method and Textual Inversion when initializing $S_\star$ with a general word (*object* or *animal*).

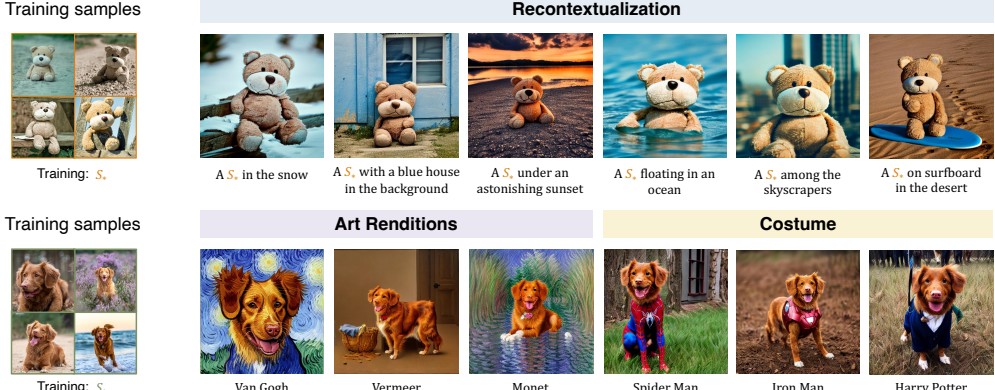

Figure 7: **Applications.** We use different contexts, artistic styles, and various costume outfits to generate images of high image fidelity and text fidelity.

**Robust $S_\star$ initialization.** Proper word initialization is crucial for Textual Inversion (Gal et al., 2023a) due to its high sensitivity to the chosen initialization word. In contrast, ViCo is robust to such initialization variations, benefiting from the visual condition. When unsure about a suitable initialization word, "object" or "animal" can be generally reliable options. Fig. 6 compares different initialization words for Textual Inversion and our method. While Textual Inversion exhibits severe distortion when initialized with "object" or "animal", our approach maintains high-quality generation.

### 4.4 APPLICATIONS

We show three types of applications of ViCo in Fig. 7. The first application is *recontextualization*. We generate images for a novel object in different contexts. The generated results present natural-looking and unobtrusive integration of the object and the contexts, with diverse poses (*e.g.*, sitting, standing, and floating). We also generate *art renditions* of novel objects in different painting styles. We use a text prompt "a painting of a $S_\star$ in the style of [painter]". Our results have novel poses that are unseen in the training samples, *e.g.*, the painting in the style of "Vermeer". In addition, we change the *costume* for the novel object using a text prompt "a $S_\star$ in a [figment] outfit", producing novel image variations while preserving the appearance of novel objects.

### 5 CONCLUSION

In summary, our paper introduces ViCo, a fast and lightweight method for personalized text-to-image generation that preserves fine object-specific details. Our approach incorporates visual conditions into the diffusion process through an image cross-attention module, enabling the extraction of accurate object masks. These masks effectively isolate the object of interest, eliminating distractions from the background in the latent space. Our visual condition module seamlessly integrates with pretrained diffusion models without the need for diffusion fine-tuning, allowing for scalable deployment. Moreover, our model is easy to use, as it doesn't rely on prior object masks or extensive preprocessing.

**Limitations.** We also notice certain limitations of our method. The decision to keep the diffusion model frozen can sometimes result in lower performance compared to methods that fine-tune the original diffusion model (Ruiz et al., 2023; Kumari et al., 2023). Additionally, the use of Otsu thresholding for mask binarization adds a slight time overhead during training and inference for each sampling step. However, these limitations are mitigated by the shorter training time, as our method requires no preprocessing and is optimized for fewer steps, and the negligible increase in inference time (several seconds), which has minimal impact on the overall model implementation.

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

## A   MORE DETAILS ON IMPLEMENTATION

**Architecture.** The proposed image cross-attention blocks, designed to accept visual conditions with the standard attention architecture as in (Vaswani et al., 2017), are integrated into specific attention layers within the decoder of the diffusion U-Net architecture. Specifically, we incorporate these blocks into every other attention layer in the decoder of the U-Net, to achieve balanced and effective performance. This design is based on the observation that integrating visual-condition attention into decoder layers produces better results compared to encoder layers, as shown in Fig. 8. We also observe that integrating visual-condition attention into both layers yields comparable performance to integrating it solely in the decoder layers. Therefore, we opt to exclusively integrate visual-conditioned attention in the decoder layers in order to reduce the parameter load and achieve a more lightweight design. The details of which attention layers are incorporated with the visual condition can be found in Tab. 5.

Table 5: **Architecture scheme.** Diffusion U-Net consists of encoder, middle, and decoder layers, with 16 original cross-attention blocks. The last row indicates the integration of visual-condition attention in specific cross-attention layers.

| U-Net | Encoder | Middle | Decoder | | | | | | | | |
|---|---|---|---|---|---|---|---|---|---|---|---|
| Attention index | 0 − 5 | 6 | 7 | 8 | 9 | 10 | 11 | 12 | 13 | 14 | 15 |
| Visual condition? | ✗ | ✗ | ✗ | ✓ | ✗ | ✓ | ✗ | ✓ | ✗ | ✓ | ✗ |

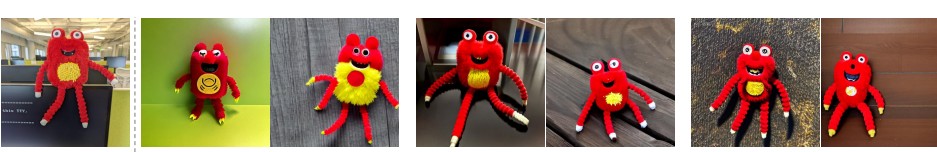

|  Sample  |  Encoder  |  Decoder (**ours**)  |  Encoder + Decoder  |

Figure 8: Comparison of integrating visual-condition attention into encoder layers, decoder layers, and both layers.

**Masking strategy.** In Fig. 9, we present a comparison between our automatic mask and the ground-truth mask generated by SAM (Kirillov et al., 2023). Additionally, we compare our masking strategy and attention alignment. The alignment is enforced by employing MSE between the $S_\star$ cross-attention map and either the ground-truth mask or our automatic mask. It is important to note

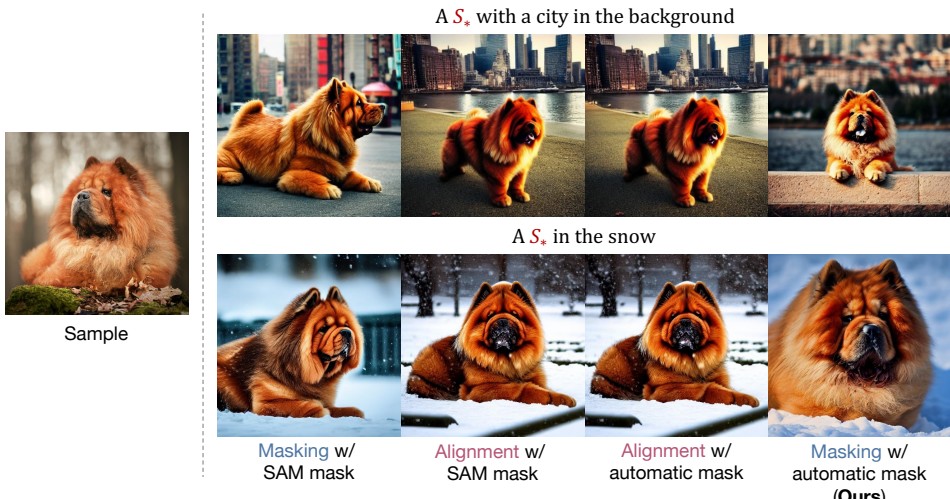

Figure 9: **Comparison of different mask settings.** We employ SAM (Kirillov et al., 2023) to generate the so-called ground-truth object mask.

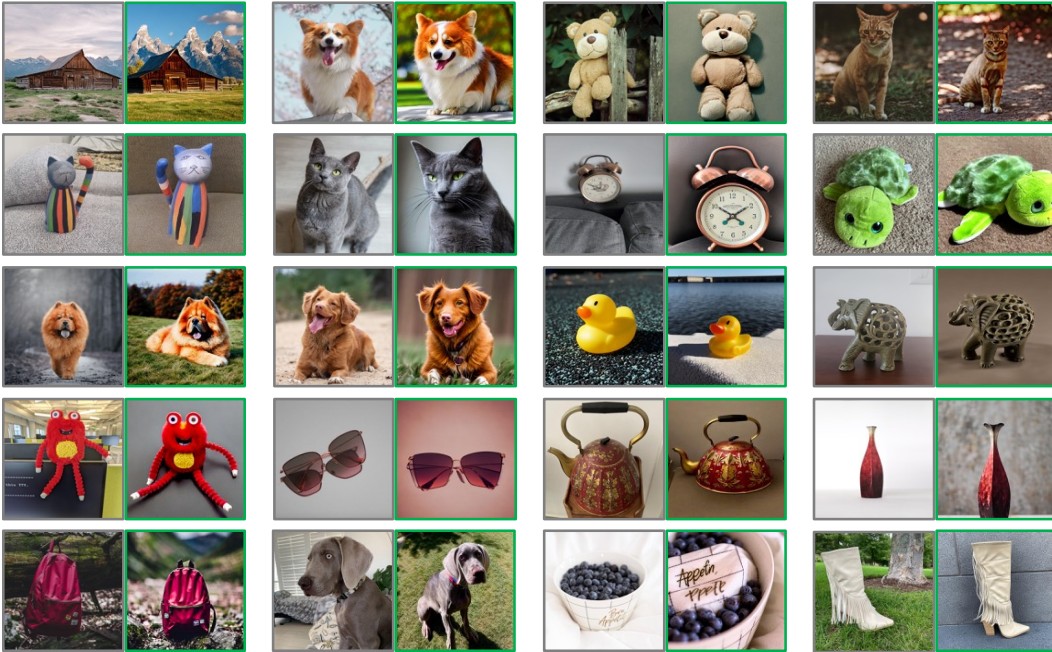

Figure 10: **Training samples and naive generations.** For each image pair, we show one training sample of a unique object on the left, and one generation result using our model with the text prompt "a photo of a $S_\star$" on the right.

that our automatic mask is inherently derived from the attention, making the alignment process akin to a self-supervised technique. Remarkably, the masking performance using the ground-truth mask is on par with our automatic mask, demonstrating the effectiveness of our automatic mask. The results from attention alignment using both masks closely resemble each other but fall short of the performance achieved using the proposed masking strategy.

**Training data sampling.** During training, we sample training images in sequential order and sample reference images randomly from the rest. This approach ensures that for each step, the training image and the reference image are different, allowing the model to focus on learning the shared novel concept between the two images rather than the entire image.

Table 6: **More information on training images.**

| Index | Object | Category | From | #Samples |
|---|---|---|---|---|
| 0 | cat statue | Toy | Textual Inversion | 6 |
| 1 | elephant statue | Toy | Textual Inversion | 5 |
| 2 | duck toy | Toy | DreamBooth | 4 |
| 3 | monster toy | Toy | DreamBooth | 5 |
| 4 | teddy bear | Toy | Custom Diffusion | 7 |
| 5 | tortoise plushy | Toy | Custom Diffusion | 12 |
| 6 | brown dog | Pet | DreamBooth | 5 |
| 7 | fat dog | Pet | DreamBooth | 6 |
| 8 | brown dog | Pet | DreamBooth | 5 |
| 9 | black cat | Pet | DreamBooth | 5 |
| 10 | brown cat | Pet | DreamBooth | 5 |
| 11 | black dog | Pet | Custom Diffusion | 8 |
| 12 | clock | Accessory | Textual Inversion | 5 |
| 13 | pink sunglasses | Accessory | DreamBooth | 6 |
| 14 | fancy boot | Accessory | DreamBooth | 6 |
| 15 | backpack | Accessory | DreamBooth | 6 |
| 16 | berry bowl | Container | DreamBooth | 6 |
| 17 | red teapot | Container | Textual Inversion | 5 |
| 18 | vase | Container | DreamBooth | 6 |
| 19 | barn | Building | Custom Diffusion | 7 |

Table 7: **Text prompt list for quantitative evaluation.** "{}" represents $S_\star$ in Textual Inversion (Gal et al., 2023a) and ours, and represents "[V] class" in DreamBooth (Ruiz et al., 2023) and Custom Diffusion (Kumari et al., 2023).

| Text prompts for non-live objects | Text prompts for live objects |
|---|---|
| "a {} in the jungle" | "a {} in the jungle" |
| "a {} in the snow" | "a {} in the snow" |
| "a {} on the beach" | "a {} on the beach" |
| "a {} on a cobblestone street" | "a {} on a cobblestone street" |
| "a {} on top of pink fabric" | "a {} on top of pink fabric" |
| "a {} on top of a wooden floor" | "a {} on top of a wooden floor" |
| "a {} with a city in the background" | "a {} with a city in the background" |
| "a {} with a mountain in the background" | "a {} with a mountain in the background" |
| "a {} with a blue house in the background" | "a {} with a blue house in the background" |
| "a {} on top of a purple rug in a forest" | "a {} on top of a purple rug in a forest" |
| "a {} with a wheat field in the background" | "a {} wearing a red hat" |
| "a {} with a tree and autumn leaves in the background" | "a {} wearing a santa hat" |
| "a {} with the Eiffel Tower in the background" | "a {} wearing a rainbow scarf" |
| "a {} floating on top of water" | "a {} wearing a black top hat and a monocle" |
| "a {} floating in an ocean of milk" | "a {} in a chef outfit" |
| "a {} on top of green grass with sunflowers around it" | "a {} in a firefighter outfit" |
| "a {} on top of a mirror" | "a {} in a police outfit" |
| "a {} on top of the sidewalk in a crowded street" | "a {} wearing pink glasses" |
| "a {} on top of a dirt road" | "a {} wearing a yellow shirt" |
| "a {} on top of a white rug" | "a {} in a purple wizard outfit" |
| "a red {}" | "a red {}" |
| "a purple {}" | "a purple {}" |
| "a shiny {}" | "a shiny {}" |
| "a wet {}" | "a wet {}" |
| "a {} with Japanese modern city street in the background" | "a {} with Japanese modern city street in the background" |
| "a {} with a landscape from the Moon" | "a {} with a landscape from the Moon" |
| "a {} among the skyscrapers in New York city" | "a {} among the skyscrapers in New York city" |
| "a {} with a beautiful sunset" | "a {} with a beautiful sunset" |
| "a {} in a movie theater" | "a {} in a movie theater" |
| "a {} in a luxurious interior living room" | "a {} in a luxurious interior living room" |
| "a {} in a dream of a distant galaxy" | "a {} in a dream of a distant galaxy" |

## B  DATASET DETAILS

**Training images.** For quantitative evaluation, our training images comprise 20 objects in 5 categories, namely 6 toys, 6 live animals, 4 accessories, 3 containers, and 1 building, selected from Textual Inversion (Gal et al., 2023a), DreamBooth (Ruiz et al., 2023), and Custom Diffusion (Kumari et al., 2023), allowing a fair and comprehensive evaluation. We list the objects and their information in Tab. 6. We show the training image samples in Fig. 10, with our naive generations, *i.e.*, generated images using "a photo of a $S_\star$". We observe that the naive generations successfully preserve the original object, demonstrating the effectiveness of our model in capturing and reproducing intricate visual details.

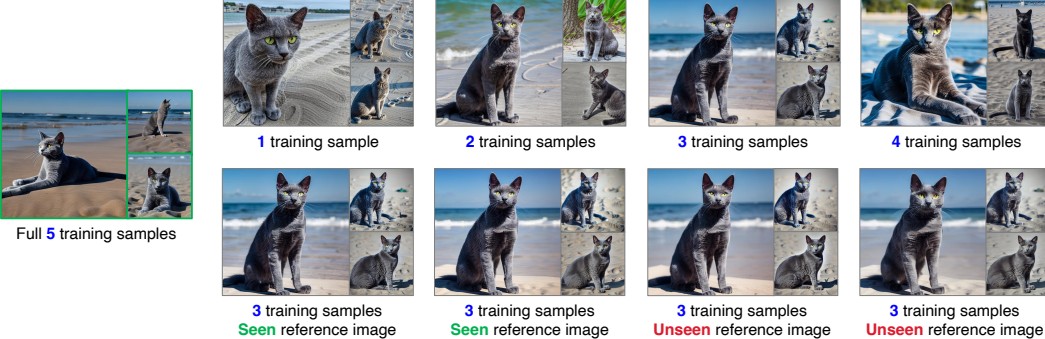

Figure 11: **Full 5 training samples *vs.* varying training samples and reference images.** On the left, we present the generated images using our model with the full 5 training samples. On the right, the top row showcases the generated images with different numbers of training samples. In the bottom row, we display the generated images using different reference images during inference, including both seen images from the training set and unseen images. This analysis provides insights into the effects of training sample size and reference image selection on the image generation process.

**Text prompts.** We adopt the same text prompt list used in Textual Inversion (Gal et al., 2023a) for training. For quantitative evaluation, we collect 31 prompts for 11 non-live objects and 31 prompts for 5 live animals in total. We show them in Tab. 7.

## C ANALYSIS OF VARYING TRAINING SAMPLES AND REFERENCE IMAGES

**The number of training samples.** In the main paper, we follow the standard training protocol outlined in (Gal et al., 2023a), where all available training samples for each object are utilized. However, we also conduct additional experiments to examine the impact of varying the number of training samples on the generation performance, as depicted in the top row of Fig. 11. Specifically, we select the object "black cat" and employ 1, 2, 3, and 4 training samples from the complete dataset of 5 training samples in total. In the case of using only 1 training sample, it is employed as both the denoising target and the reference image. The generated images from scenarios with only 1 or 2 training samples exhibit a tendency to overfit the object, resulting in a diminished representation of the textual information. Nevertheless, across all cases, our method consistently demonstrates high image fidelity and quality.

**Different reference images.** In addition, we also evaluate the generation performance by employing different reference images during inference, including both seen images from the training set and unseen ones. The results are presented in the bottom row of Fig. 11. Remarkably, we observe that the variations in the generated images are minimal when using the same random seed, underscoring the robustness of our proposed visual condition to the input image. This finding suggests that our model can effectively generalize and maintain consistent performance regardless of whether the reference image is seen or unseen during training.

**Multiple reference images.** Our image cross-attention mechanism has the ability to handle a variable number of input tokens. This flexibility enables us to seamlessly use multiple reference images by accommodating any number of tokens from the concatenated reference images. In Fig. 12, we compare the results obtained by using two reference images with those obtained by using a single reference image. We observe that using different types of reference images produces similar generated images, which highlights the robustness of the image cross-attention mechanism.

## D TRAINING STEP DISCUSSION

In the main paper, we report all results at the checkpoint of 400 training steps, which generally yield the best overall performance within a short training time (∼5 minutes). However, we observe that for some objects, text information is better preserved with fewer training steps. In Fig. 13, we present some cases where training for 300 steps results in better preservation of text-related information. This observation can be attributed to the fact that more training steps can potentially lead to overfitting the generation to the training images, thereby neglecting the information provided by the text prompt to some extent. In practical applications, it is advisable to save multiple checkpoints at different training steps, allowing users to choose the most suitable checkpoint for text-prompted inference.

A $S_*$ on the beach

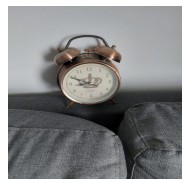 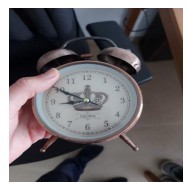 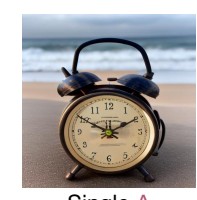 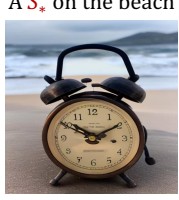 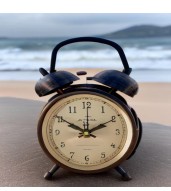

Reference A          Reference B          Single A          Single B          A + B

Figure 12: **Multiple reference images *vs*. a single reference image.** For multiple reference images, we concatenate the tokens corresponding to each image and pass the concatenated tokens through the image cross-attention.

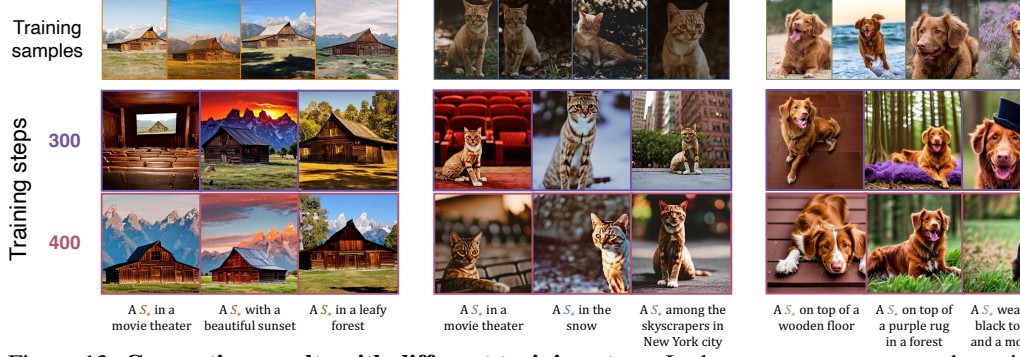

Figure 13: **Generation results with different training steps.** In the top row, we present the training samples. The middle row showcases the generated images after training for 300 steps, while the bottom row displays the generated images after training for 400 steps. Each object is evaluated using three distinct text prompts, ensuring a comprehensive and unbiased assessment.

## E  USER STUDY

To gain insights into human preferences regarding generation performance, we conduct a user study to evaluate our model along with the compared methods. The study consists of 18 samples of different objects, including some objects used for quantitative evaluation as well as additional new ones. These samples are prompted by various text prompts.

During each trial, we generated 8 images using each method and selected the most visually appealing and best-aligned image as the candidate for comparison. For each question in the study, users were asked to assess the image candidates and choose the best one (or two if they found two results equally good) based on three perspectives: image quality, text fidelity, and object fidelity. An example question is shown in Fig. 14. In total, we collected answers from 40 users for a total of 18 comparative questions, resulting in 720 individual responses across the three evaluation metrics. The user study votes are plotted in Fig. 15 and the percentage results of the votes are reported in Tab. 8, providing a distribution summary of the user preferences for each evaluated metric.

The results indicate that our method significantly outperforms the other models in all metrics. Note that the users involved in the study have no prior knowledge of the specific task details and many of them come from non-technical backgrounds. The evaluation metrics used in the user study may be subjective in nature, as they rely on the personal opinions and preferences of the participants. However, this subjective evaluation provides a more human-centered perspective, which ensures that our model produces outputs that align with human preferences and expectations.

Table 8: **Comparison of user preferences.** The percentages of votes are reported.

|  | Image quality | Text fidelity | Object fidelity |
|---|---|---|---|
| Textual Inversion (Gal et al., 2023a) | 0.117 | 0.054 | 0.076 |
| Custom Diffusion (Kumari et al., 2023) | 0.259 | 0.288 | 0.158 |
| DreamBooth (Ruiz et al., 2023) | 0.249 | 0.301 | 0.318 |
| ViCo (ours) | **0.375** | **0.357** | **0.448** |

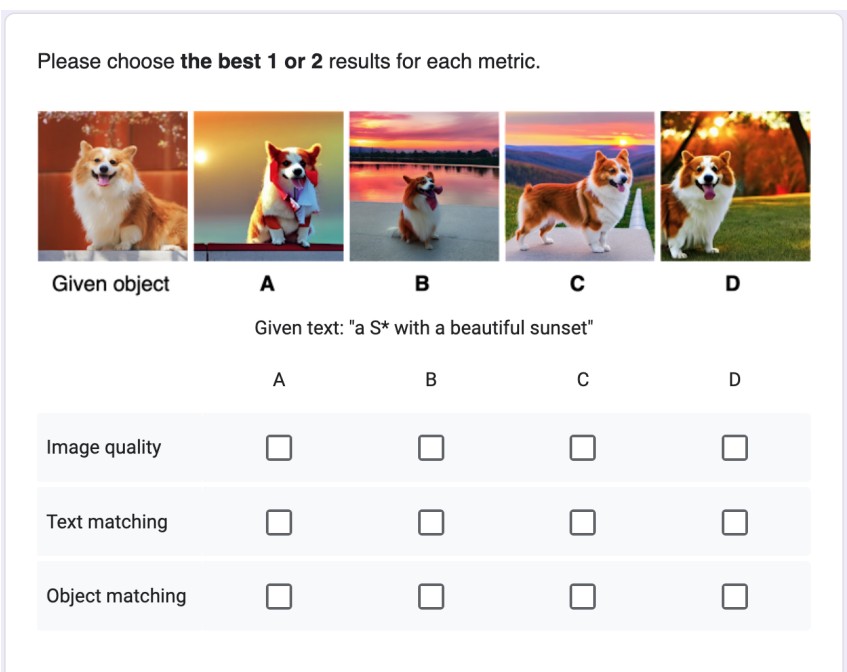

Figure 14: **An example question of the user study.** We provide the reference image and the text prompt and ask the users to vote for one or two candidates based on three metrics.

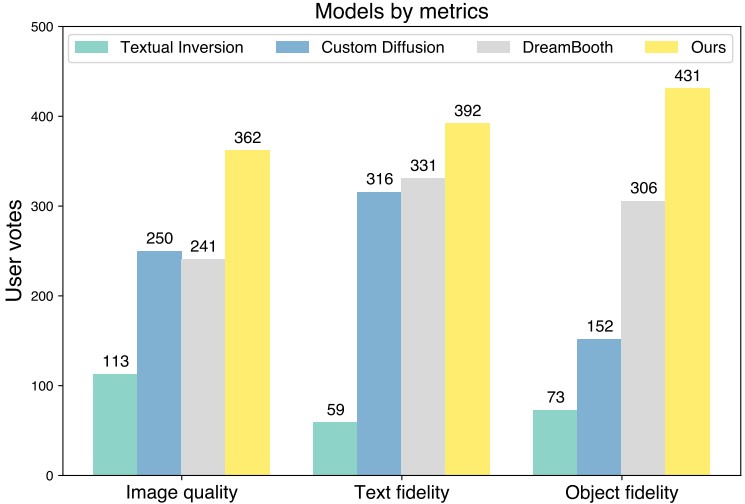

Figure 15: **User votes in three metrics.** Note that the sum of votes in each metric is more than 720 because the user can vote for 2 candidates in one question if they find two results equally good.

## F  FAILURE CASES

In Fig. 16, we present several failure cases encountered by our model, highlighting the challenges and areas for improvement. These failure cases can be categorized into two types: image misalignment and text misalignment. Image misalignment occurs when the object has an intricate appearance, making it difficult for the model to accurately capture and reproduce all the details. Additionally, image misalignment can also occur when there are multiple objects of the same category that co-occur, leading to difficulties in properly distinguishing the individual object of interest. Text misalignment, on the other hand, is primarily caused by two factors. Firstly, it can occur due to the loss of text information, resulting in a mismatch between the intended text prompt and the synthesized image. Secondly, text misalignment can arise from the undesirable synthesis of the object and the text prompt, leading to unexpected or nonsensical combinations. While our current model faces these challenges, we acknowledge them as areas for future improvement and research.

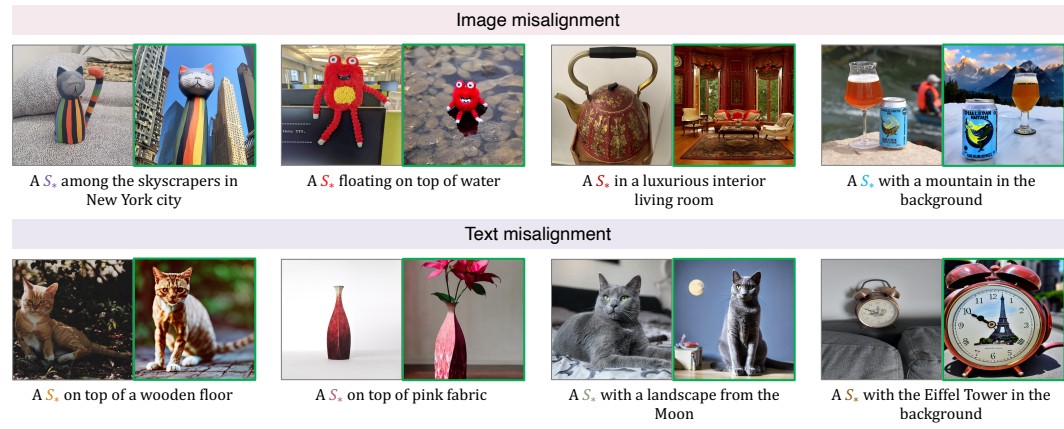

Figure 16: **Failure cases.** We present two types of failure cases: image misalignment and text misalignment. Each pair of samples consists of a reference image on the left and the corresponding generated image on the right.

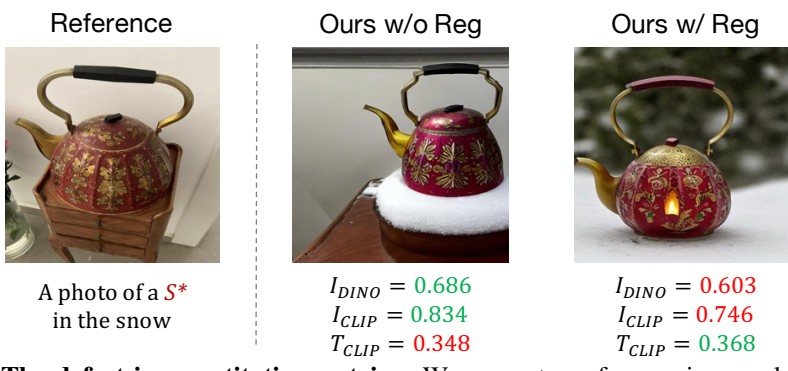

Figure 17: **The defect in quantitative metrics.** We present a reference image along with two generated images: one produced with the regularization applied and the other without it. We compute the three quantitative metrics below each generated image.

## G  DEFECT IN QUANTITATIVE METRICS

Our quantitative metrics are based on pretrained models CLIP (Radford et al., 2021) and DINO (Caron et al., 2021), which produce global representations for images. Therefore, it may be hard to reflect some local details in quantitative comparison stemming from comparing feature vectors. For example, we present two images generated with and without the regularization and compute three similarity scores in Fig. 17. We notice that although the image generated using the regularization shows better quality, its image similarity metric greatly lags behind the other because the one without the regularization overfits to the training data to some extent (presenting the "cabinet"). We believe a more proper evaluation metric that can reflect the details in the images is desired for future advancement.

## H  ADDITIONAL VISUALIZATIONS OF ATTENTION

**Mask visualization.**  We visualize the mask of the reference image derived from the attention map during training and inference. In Fig. 18, we visualize attention maps and corresponding binarized masks along with sampling steps, which exhibit the notable effect of image matting. In Fig. 19, we visualize the object mask throughout the training process. In the initial steps, the mask rapidly converges to a reliable indicator for object segmentation, consistently showcasing the significant image matting in subsequent steps.

**Effect of the regularization on attention.**  We provide additional visualization results to show that regularization effectively directs the attention to focus on the object of interest in Fig. 20. We also observed that in certain cases, such as the last row in Fig. 20, the regularization had minimal impact on the attention. Therefore, in practical applications, users have the flexibility to customize

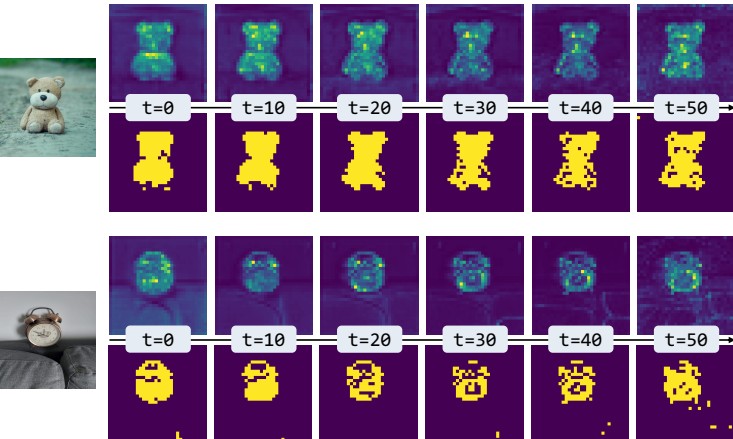

Figure 18: **Mask samples in inference.** For each instance ("teddy bear" and "clock"), given a reference image (left), we visualize the attention associated with $S_\star$ (top) and the corresponding mask (bottom) of it with an interval of 10 during the 50 inference steps.

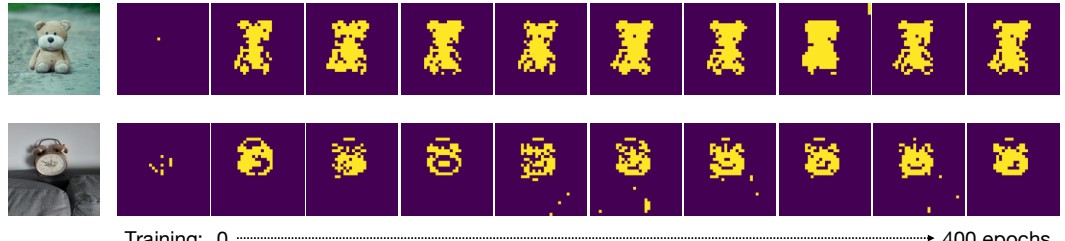

Figure 19: **Mask samples during training.** Given a reference image (left), we visualize the mask of it, uniformly sampled throughout the training process. Except for the mask at the very beginning of training, all masks in subsequent steps present good image matting, which can be efficiently used in the visual condition module.

the weight of the regularization during training to further control the attention behavior according to their specific needs.

**Attention at each step.** The visualized attention presented in Fig. 20 represents the average attention across all inference steps. Additionally, we provide visualizations of the attention at each inference step with an interval of 5 in Fig. 21, allowing for a more detailed observation of the attention dynamics throughout the generation process. In the early steps, the attention tends to exhibit more scattered responses, exploring different regions in the generated image. As the inference progresses, the attention quickly converges and becomes more focused on the object of interest.

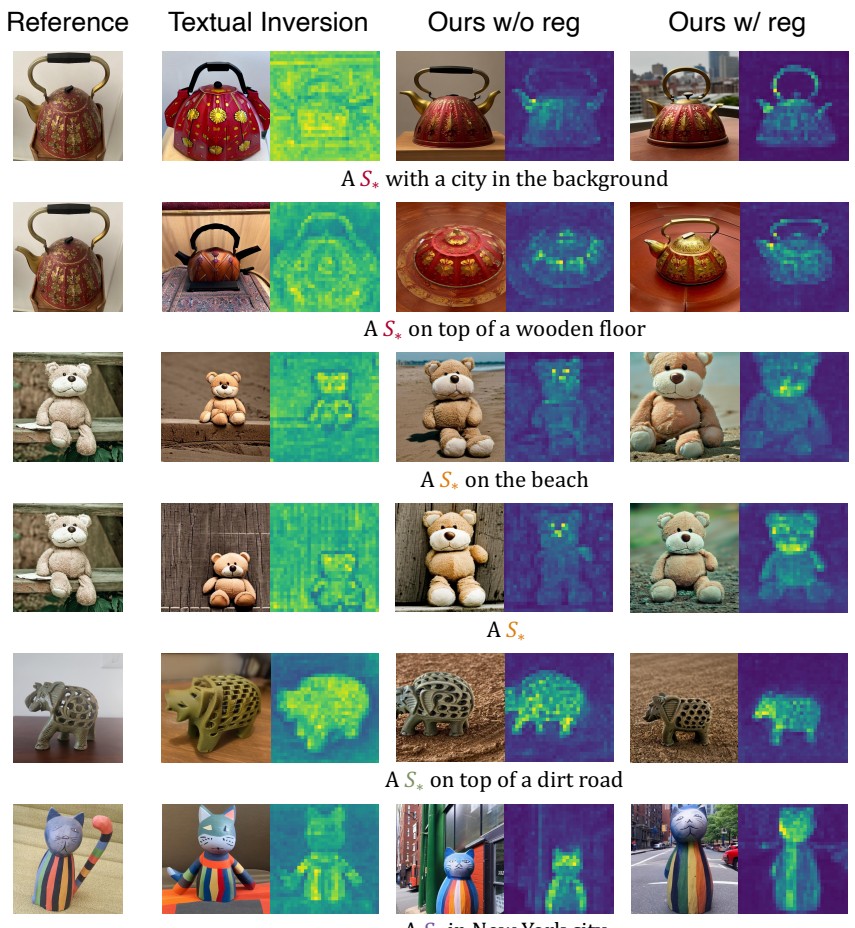

Figure 20: **Visualization of attention associated with $S_\star$.** We visualize the average attention in the inference process of Textual Inversion (Gal et al., 2023a) and ours with or without the proposed regularization.

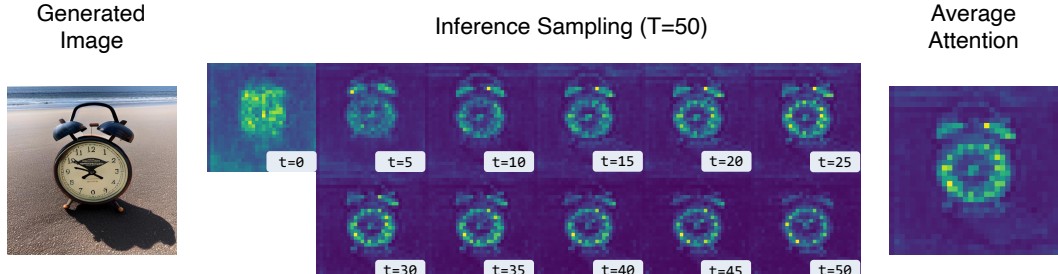

Figure 21: **Inference steps.** We visualize the attention associated with $S_\star$ with an interval of 5 during the 50 inference steps. We also present the generated image (on the left) and the average attention along with all inference steps (on the right).

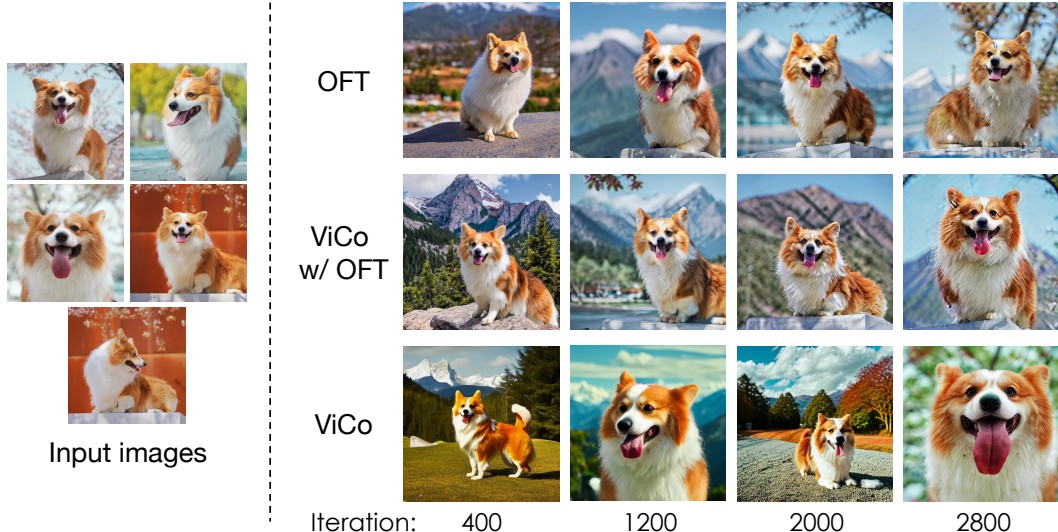

Figure 22: **Generation results across different number of iterations.** We compare ViCo, OFT (Qiu et al., 2023), and using both ViCo and OFT when training different iterations.

## I  ADDITIONAL COMPARISON RESULTS

Due to the page limit of the main paper, we include additional generation results in Fig. 26. The results compare our model with Textual Inversion (Gal et al., 2023a), DreamBooth (Ruiz et al., 2023), and Custom Diffusion (Kumari et al., 2023). For each comparison, we select the visually best image from a set of 8 randomly generated images using different objects. We observe that our method performs on par with, or even surpasses, the finetuning-based methods DreamBooth (Ruiz et al., 2023) and Custom Diffusion (Kumari et al., 2023). The generated images exhibit high quality in terms of image fidelity, text fidelity, text-image equilibrium, authenticity, and diversity.

**Comparison with OFT.**    Orthogonal Finetuning (OFT) (Qiu et al., 2023) is a recently proposed fine-tuning method that can be efficiently used to fine-tune DreamBooth (Ruiz et al., 2023). OFT demonstrates stability in generating images even with a large number of training steps. It can also be applied in conjunction with ViCo. In Fig. 22, we present the generated results comparing different iterations of OFT, ViCo, and ViCo with OFT. In the early steps (e.g., 400), ViCo is capable of producing images that preserve fine object details, while OFT has not yet learned accurate object-specific semantics. As the number of training steps increases significantly (e.g., 2800), the generated images by OFT exhibit slight distortions, and ViCo may overfit to the training image, resulting in the loss of text information. By combining ViCo with OFT, we can address both of these issues and achieve the highest generation quality at all iterations.

**Comparison with the encoder-based method ET4.**    Encoder for Tuning (E4T) (Gal et al., 2023b) is a representative work of encoder-based models (Shi et al., 2023; Jia et al., 2023; Gal et al., 2023b; Chen et al., 2023). It is worth noting that these models, including E4T, are not directly related to our current study, as they necessitate training on extensive datasets specific to particular category domains. In Fig. 23, we compare our model with E4T which is pretrained on a substantial dataset of human faces. According to Gal et al. (2023b), a large batch size of 16 or larger is crucial. To adhere to this setting, we implement the use of gradient accumulation over 16 steps in E4T. Notably, our model excels in preserving facial details compared to E4T.

## J  ADDITIONAL RESULTS OF VARIOUS APPLICATIONS

We present the results of our model deployed to various applications in Fig. 27. In addition to the applications of recontextualization, art renditions, and costume changing, which have been showcased in the main paper, we further demonstrate the effectiveness of our model in *activity control*, such as controlling a dog to sleep or to jump, and *attribute editing*, such as editing the pattern or the color of a container. We also show more generated results in terms of complicated style change in Fig. 28, highlighting our model's edit-ability to effectively condition on complex styles. Furthermore, we

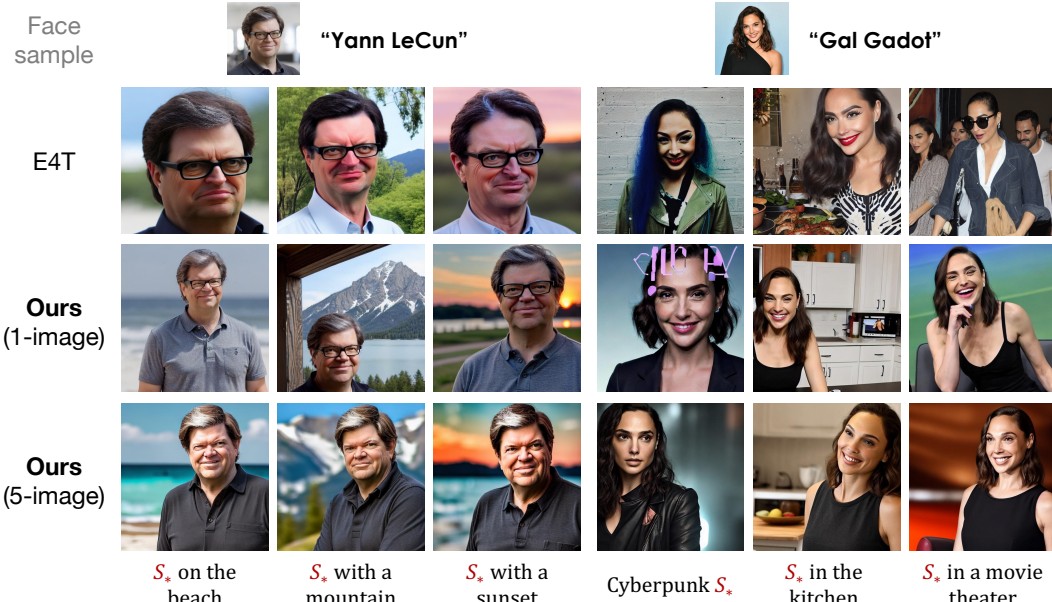

Figure 23: **Comparison with E4T (Gal et al., 2023b) on face images**. E4T is initially pretrained on a large-scale dataset of human faces and then fine-tuned using a single image of "Yann Lecun" or "Gal Gadot". We present our results by training `ViCo` on either a single image or five images.

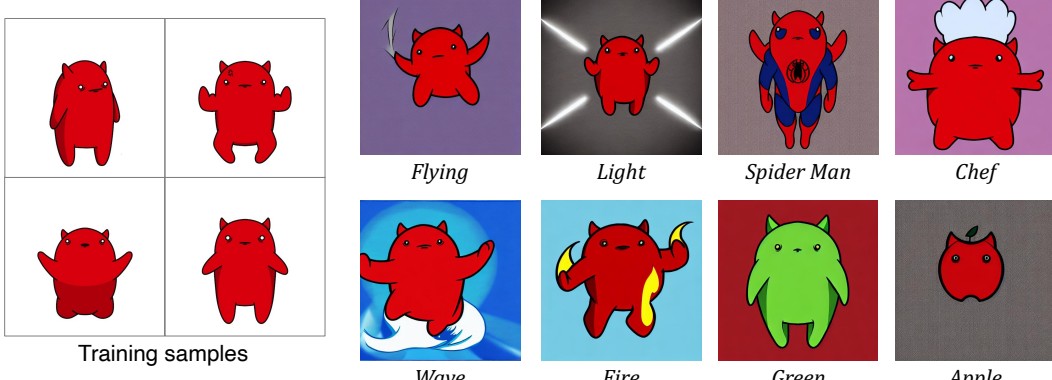

Figure 24: **Comic character generation.** We generate text-guided images (on the right) with the given comic object (on the left). Our results preserve the comic style and the appearance of the object.

present the implementation of our model for *comic character generation* in Fig. 24. This application allows us to generate comic-style images that exhibit a text-guided property. These additional examples highlight the versatility and flexibility of our model in various creative and interactive scenarios.

**Multi-object composition.** Our model can also be easily altered to support multi-object composition with two different objects. Particularly, we train $S1_\star$, $S2_\star$, and unified image cross-attention blocks with two datasets of different objects. In inference, we feed two different reference images into the image cross-attention by token concatenation and respectively obtain object masks from $S1_\star$, $S2_\star$. We report our results of multi-object composition in Fig. 25.

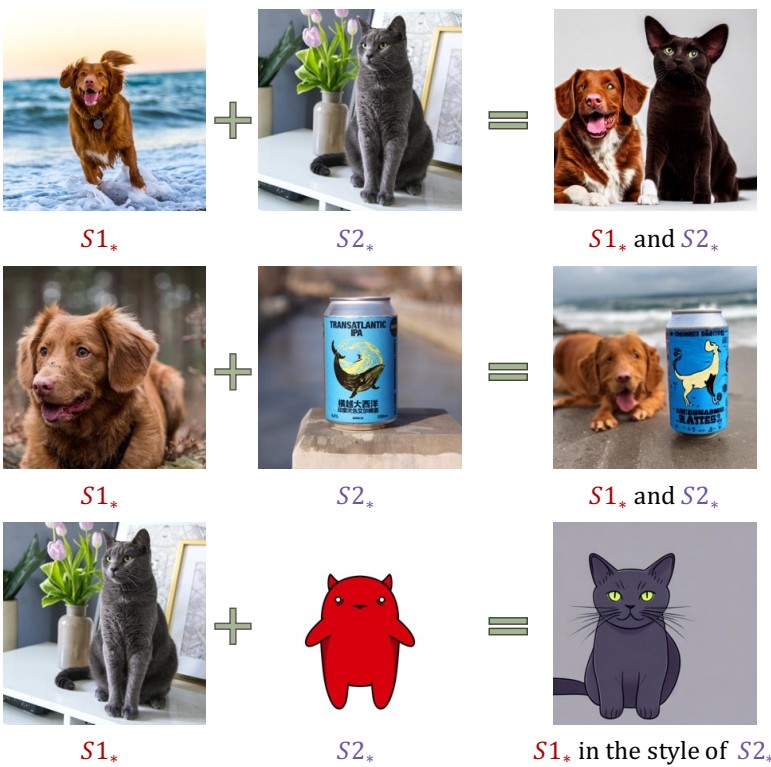

Figure 25: **Multi-object composition.** Our model supports multi-object compositions, demonstrating results in two composition types: (1) simultaneous appearance of two objects, and (2) one object in the style of the other object.

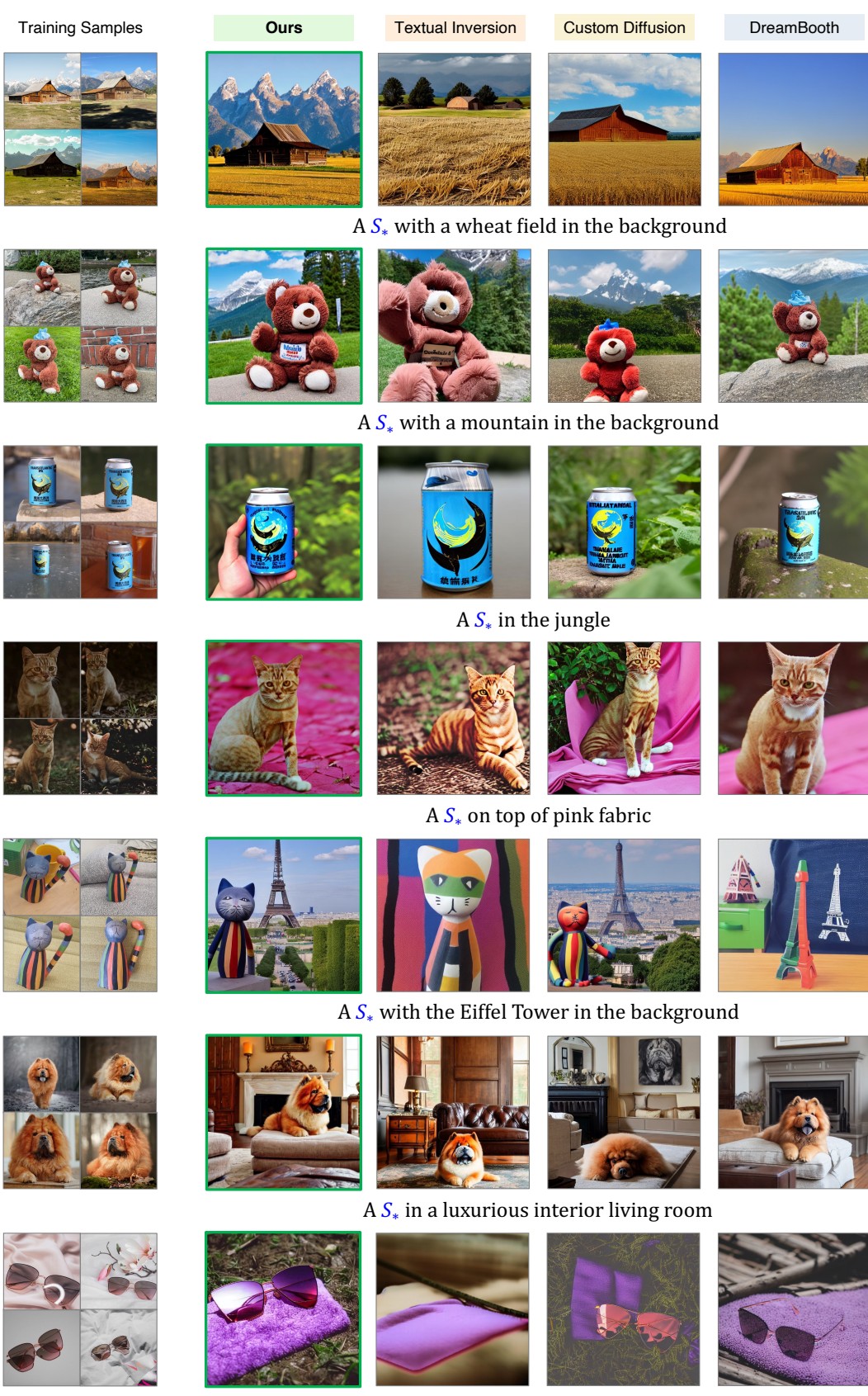

Figure 26: **Additional comparison results on more objects.** Our model consistently demonstrates superb performance across all experimental trials. Zoom in to see the image details.

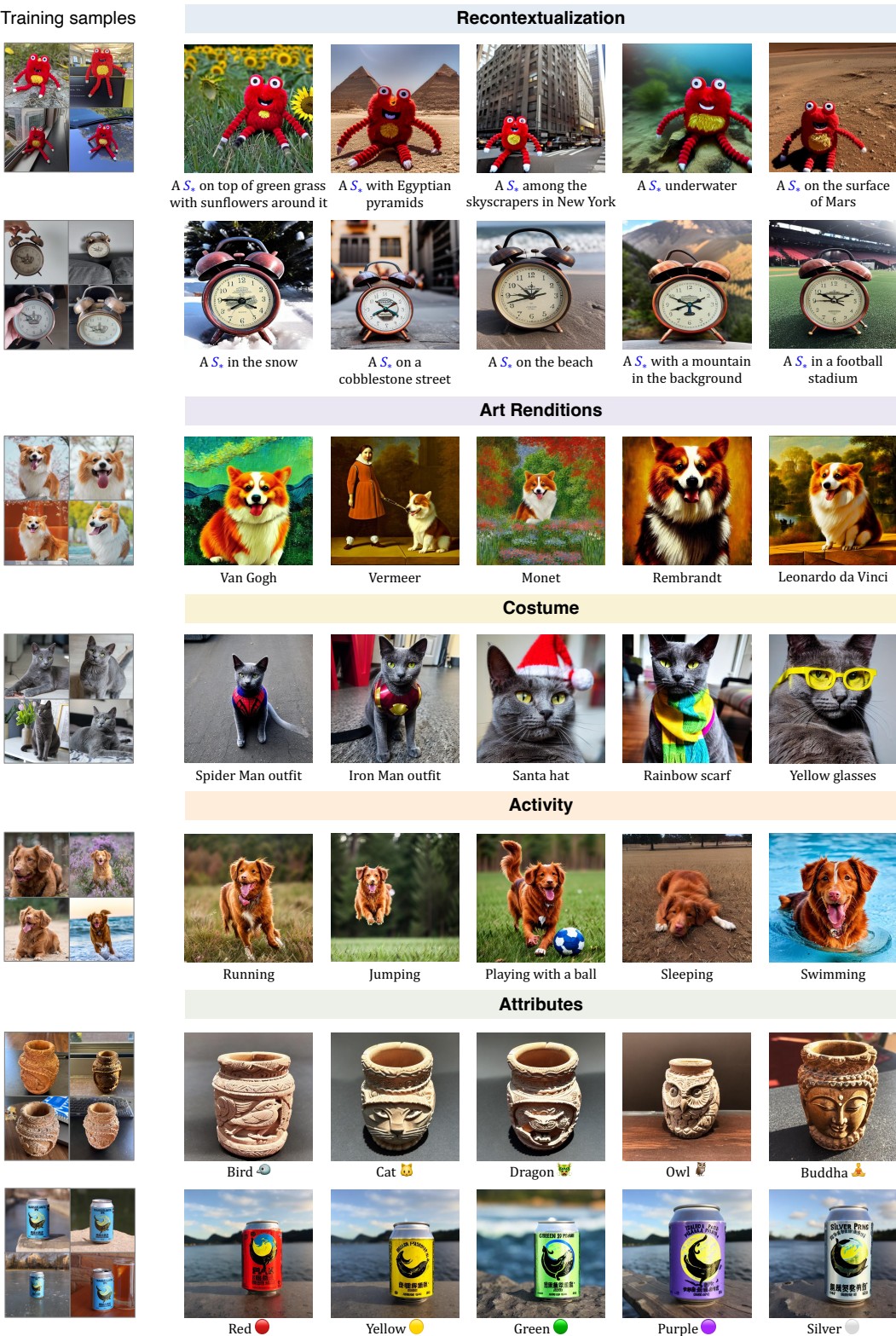

Figure 27: **Additional applications using our method.** `ViCo` excels in various tasks, including recontextualizing input images, synthesizing diverse art renditions, changing costumes, controlling object poses in different activities, and editing object intrinsic attributes. The generated images exhibit high authenticity. Zoom in to see the image details.

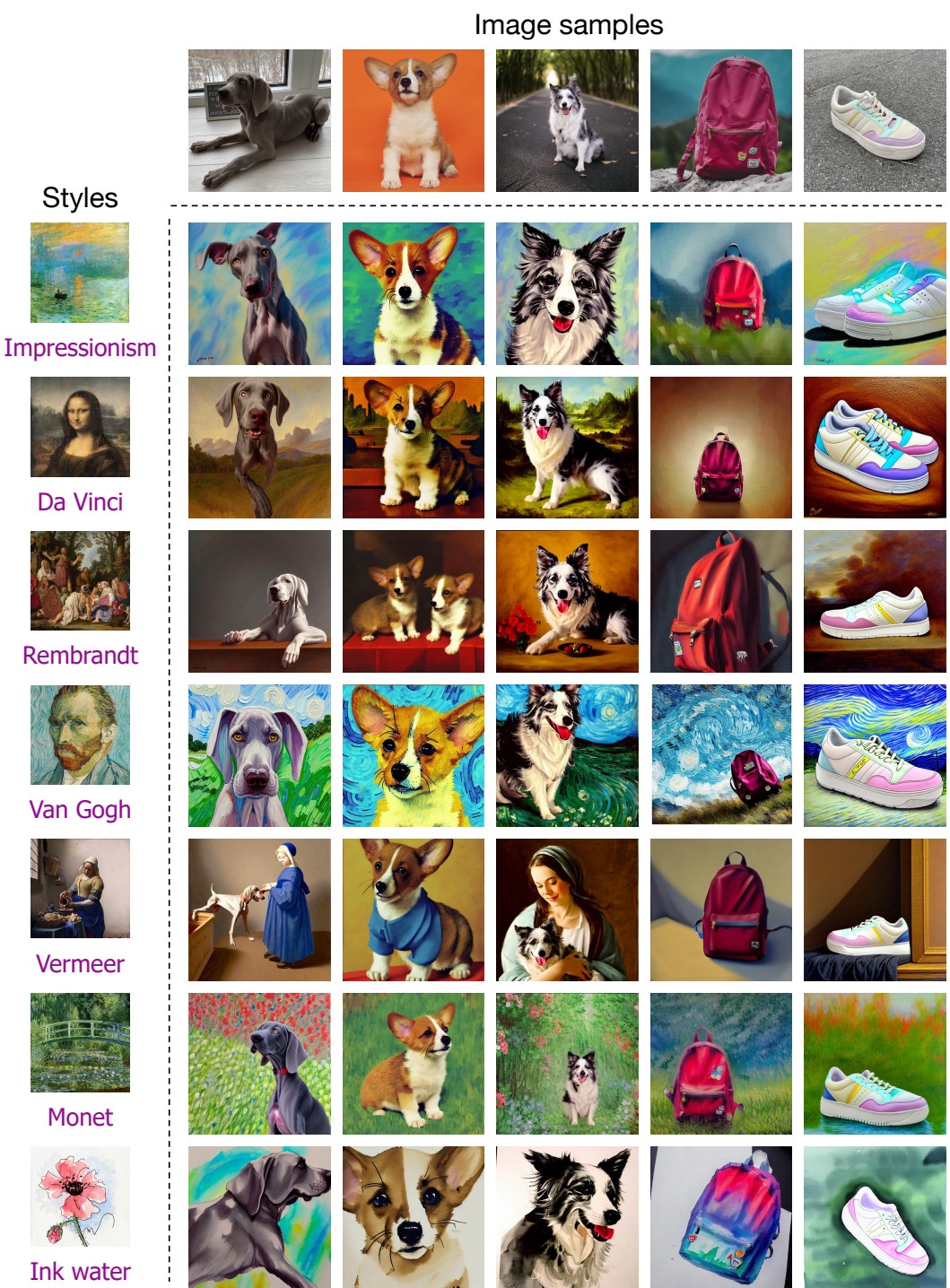

Figure 28: **Generated results in various styles.** Each style is prompted by the text condition "a painting of $S_\star$ in the style of [STYLE]".

# K    BROADER IMPACTS

Our research on personalized text-to-image generation has significant impacts, both positive and negative. On the positive side, our approach has numerous applications, such as recontextualization, art renditions, and costume changing, which can be widely utilized in industries like artistic creations, media production, and advertising. For example, in advertising, designers can efficiently synthesize drafts of the advertising subject and the desired context with minimal effort.

However, we acknowledge that our research also raises social concerns regarding the potential unethical use of fake images. There is a risk of malicious exploitation as our model allows easy generation of photorealistic images by anyone, even without prior technical knowledge. This could lead to the fabrication of images that include specific individuals or personal belongings. For instance, using readily available selfies from the internet, individuals with malicious intent could fabricate images to engage in fraudulent activities or defamatory actions by portraying someone unethically.

To mitigate the potential negative impacts, we strongly advocate for strict regulation and responsible use of this technology. It is essential to establish guidelines and ethical frameworks to govern the deployment and application of personalized text-to-image generation models like ours. With proper regulations, we can help prevent misuse and ensure that this technology is used for legitimate and ethical purposes. This could include measures such as obtaining consent for image generation, implementing authentication mechanisms, and promoting public awareness about the risks and ethical considerations associated with this technology.

