# OpenReview forum: "ViCo: Plug-and-play Visual Condition for Personalized Text-to-image Generation"
_ICLR.cc/2024/Conference — Submitted to ICLR 2024_

### Official Review · Reviewer_Z7sX · 2023-10-23

**Soundness:** 3 good
**Presentation:** 3 good
**Contribution:** 3 good
**Rating:** 6
**Confidence:** 3

**Summary:**

The paper presents ViCo, a novel method for personalized text-to-image generation using diffusion models. This task aims to generate photorealistic images from textual descriptions without fine-tuning the original diffusion model. It utilizes an image attention module and a mask to condition the diffusion process on visual semantics. It outperforms existing models with minimal training, making it a promising solution for personalized text-to-image generation without the need for fine-tuning diffusion models.

**Strengths:**

1. This paper presents an efficient mechanism to generate object masks without relying on prior annotations, simplifying foreground object isolation from the background.

2. It is computationally efficient and non-parametric, reducing the influence of distracting backgrounds in training samples.

3. ViCo is highly flexible and easy to deploy, as it doesn't require fine-tuning of the original diffusion model.

4. The model requires no heavy preprocessing or mask annotations, making it easy to implement and use.

**Weaknesses:**

1. ViCo may have lower performance compared to fine-tuned methods, implying an ease-of-use vs. performance trade-off.

2. The use of Otsu thresholding for mask binarization may slightly increase time overhead during training and inference for each sampling step. However, this is offset by shorter training time and a negligible increase in inference time.

**Questions:**

1. Can you explain in more detail how ViCo generates object masks and incorporates them into the denoising process?

2. How does ViCo compare to other methods in terms of computational efficiency and parameter requirements?

3. Can you explain how ViCo achieves diverse poses and appearances in recontextualization and art renditions?

4. Can you clarify the limitations mentioned and how the trade-off between keeping the frozen diffusion model and not fine-tuning it affects performance?

5. How does ViCo's cross-attention method for capturing object-specific semantics differ from others, and what are its advantages?

---

> ### Author Response · Authors · 2023-11-22
> **Response to Reviewer Z7sX (part 1/2)**
>
> We are grateful for your valuable comments! Below is our response.
>
> >W1: ViCo may have lower performance compared to fine-tuned methods, implying an ease-of-use vs. performance trade-off.
>
> Our model offers a **plug-and-play module** capable of markedly enhancing the performance of Textual Inversion, achieving results comparable to DreamBooth. It is also *feasible* to combine ViCo with other methods, such as OFT [1], which could potentially lead to further performance improvements, as demonstrated in the added experiment presented **in Figure 22 in Appendix I.** While it is technically feasible to combine ViCo with DreamBooth, we do not recommend this approach as it may compromise the flexible deployment property inherent to ViCo.
>
> >W2: The use of Otsu thresholding for mask binarization may slightly increase time overhead during training and inference for each sampling step. However, this is offset by shorter training time and a negligible increase in inference time.
>
> We can also use other *non-iterative* binarization methods to produce the mask, which can decrease the overhead to nearly zero. We choose otsu because it is robust to all types of data and consistently produces good binary masks. As the reviewer recognizes, the overhead is offset by **shorter training time** and a *negligible* increase in inference time.
>
> [1] Qiu, Zeju, et al. “Controlling text-to-image diffusion by orthogonal finetuning.” In NeurIPS, 2023.

---

> ### Author Response · Authors · 2023-11-22
> **Response to Reviewer Z7sX (part 2/2)**
>
> >Q1: Can you explain in more detail how ViCo generates object masks and incorporates them into the denoising process?
>
> We would like to explain more details as follows:
>
> - **Mask generation** takes place in the computation of each text cross-attention layer between the latent feature of the reference image and the text prompt. The cross-attention map corresponding to S*, which provides a reliable indicator of object localization, is **binarized** into an object mask, sharing the same shape as the latent shape in the current layer.
>
> - **Mask incorporation** takes place in the processing of each image cross-attention between the latent feature of the source image and the reference image. When computing $Attn = QK^T/\sqrt{d_k}$, we **mask** those elements corresponding to the masked regions of the reference image in $Attn$. In this way, the attention activations related to non-object regions in the reference image can be **suppressed**.
>
> >Q2: How does ViCo compare to other methods in terms of computational efficiency and parameter requirements?
>
> Regarding **computational efficiency**, we showed the comparison of the **time cost** in **Table 3 in Section 4.1** and report the **maximum GPU memory** used during training for all compared models:
>
> |                              | Max. GPU Mem. |
> |------------------------------|:--------:|
> | DreamBooth (optimized)       |   14.9G  |
> | Custom Diffusion |   18.3G  |
> | Textual Inversion            |   21.6G  |
> | Ours                         |   23.4G  |
>
> Note that we employ the *optimized* implementation of DreamBooth using mixed precision and 8bit Adam, to accommodate our GPUs. Without these optimizations, the training process would require over 30GB of GPU memory. During inference, all methods use approximately 19GB of GPU memory.
>
> Regarding **parameter requirements**, we reported the comparison of model parameters in **Table 1 in Section 2**.
>
> >Q3: Can you explain how ViCo achieves diverse poses and appearances in recontextualization and art renditions?
>
> The pretrained diffusion U-Net provides **rich object poses and appearance knowledge**. The rich knowledge in the pretrained diffusion model diversely guides how the object interacts with different background contexts with different poses (like "on the surfboard" and "in an ocean") in **Figure 7 in Section 4.4**. The pretrained text encoder and text cross-attention also possess **knowledge of diverse art styles**, which can transfer the raw object appearance into a specific art style without changing the object characteristics.
>
> >Q4: Can you clarify the limitations mentioned and how the trade-off between keeping the frozen diffusion model and not fine-tuning it affects performance?
>
> Finetuning the diffusion model may affect the performance in two aspects. *(1)* First, finetuning diffusion models allows more parameters to learn the object, better preserving the object details. *(2)* Second, solely training the S* token can render it dominate the text space and weaken the influence of text-related information. These effects can be demonstrated in the comparison of image similarity and text similarity metrics between Textual Inversion and DreamBooth. Based on textual inversion, our method uses a **lightweight module** to overcome the above defects but there may still be room for performance improvement.
>
> Not finetuning the diffusion model offers the advantage of **flexible transferrable deployment**. This allows for only saving a set of *small* parameters for each object without the need to save the entire diffusion model. It also facilitates *easy* transfer to other diffusion models without the requirement of re-tuning the whole model. Additionally, this approach addresses the *language drift* issue associated with finetuning the entire model as described in DreamBooth, because all established language knowledge is retained in the frozen pretrained diffusion model.
>
> >Q5: How does ViCo's cross-attention method for capturing object-specific semantics differ from others, and what are its advantages?
>
> Compared to *Textual inversion*, ViCo uses the visual condition module to incorporate **explicit visual information** that captures more object semantic details from the reference image. Our advantage is that the visual condition module can provide more information on object details than using a single token.
>
> Compared to *DreamBooth* and *Custom Diffusion*, they fine-tune the U-Net to directly learn object semantics back in the backbone model while we introduce a **plug-and-play module** to explicitly extract visual semantics and inject it into the frozen denoising process. Our advantage lies in **flexible deployment** and avoiding the *language drift* issue in DreamBooth, by keeping the diffusion model frozen.

---

### Official Review · Reviewer_dE3Q · 2023-10-30

**Soundness:** 2 fair
**Presentation:** 2 fair
**Contribution:** 2 fair
**Rating:** 5
**Confidence:** 3

**Summary:**

This work aims to achieve personalized text-to-image generation that allows users to combine inputs of texts with example images and generates an image accordingly. Existing work on this task is either computationally inefficient or sacrifices the generation quality with computation cost. The authors propose a cross-attention based mechanism, which also has the benefit of helping isolate the foreground object using attention maps, that requires only training the cross-attention layers and optimizing the placeholder embedding. The proposed method is more efficient without sacrificing the reference object's identity.

**Strengths:**

The goal of this paper is to achieve personalized text-to-image generation that is lighter-weight (and faster) than existing methods with an on par quality or even better. Quantitative and qualitative results presented seem to support this.

**Weaknesses:**

How the proposed method avoids fine-tuning the entire diffusion model for each reference object is by using cross-attention: multi-resolution features maps of the reference image, C_I^l, are used to perform cross-attention with the intermediate outputs, n_t^l, of the main denoising UNet. With cross-attention blocks, they only train these blocks for each reference object, instead of the entire diffusion model. Using cross-attention blocks in conditioned image generation to warp a source image to target [i, iv, v, vi, vii] or to preserve a reference image's identity [ii, iii] has been a popular approach. Indeed, this may be one of the first work to explore cross-attention blocks in LDMs, but I don't think this contribution seems sufficiently novel.


[i] Bhunia, Ankan Kumar, et al. "Person image synthesis via denoising diffusion model." Proceedings of the IEEE/CVF Conference on Computer Vision and Pattern Recognition. 2023.

[ii] Zhu, Luyang, et al. "TryOnDiffusion: A Tale of Two UNets." Proceedings of the IEEE/CVF Conference on Computer Vision and Pattern Recognition. 2023.

[iii] Karras, J., Holynski, A., Wang, T. C., & Kemelmacher-Shlizerman, I. (2023). Dreampose: Fashion image-to-video synthesis via stable diffusion. arXiv preprint arXiv:2304.06025.

[iv] Mallya, Arun, Ting-Chun Wang, and Ming-Yu Liu. "Implicit warping for animation with image sets." Advances in Neural Information Processing Systems 35 (2022): 22438-22450.

[v] Liu, Songhua, et al. "Dynast: Dynamic sparse transformer for exemplar-guided image generation." European Conference on Computer Vision. Cham: Springer Nature Switzerland, 2022.

[vi] Ren, Yurui, et al. "Deep image spatial transformation for person image generation." Proceedings of the IEEE/CVF Conference on Computer Vision and Pattern Recognition. 2020.

[vii] Tseng, Hung-Yu, et al. "Consistent View Synthesis with Pose-Guided Diffusion Models." Proceedings of the IEEE/CVF Conference on Computer Vision and Pattern Recognition. 2023.

**Questions:**

After reading both the main paper and supplementary sections, I'm still not 100% clear of the training procedure, which also raises questions for me regarding the results presented. Currently my understanding is that training is conducted for each object separately (in Sec.3.4 it reads "We train our model on 4-7 images with vanilla diffusion U-Net frozen..."), and during training, the learning of cross-attention layers and placeholder text embeddings S* are performed simultaneously (as described in Sec.3.4). If my understanding is correct, the question I have is: would the model learn better if cross-attention layers are trained with all available images (from all objects, or with any larger dataset where there are many objects, each with at least 2 images), and S* is optimized for each object? I have this question because when looking at the qualitative results, I think the preservation of the reference image's identity could be further improved, e.g., in Figure 1, the Batman toy's body pose changed in Figure 1, and in Figure 4, the cat statue's face changed, the texts on the can changed, and the drawing on the clock also changed. One possibility I can think of for why the reference image's identity is not perfect is that the cross-attention layers are not fully trained, and training with a larger dataset with a wider variety of objects may help.

Another question I have is regarding where to incorporate cross-attention blocks. From Supplementary Section A, it mentions that the final design was incorporating cross-attention blocks in the decoder of the UNet. Existing work  [i, ii] that also use cross-attention incorporate it in both the encoder and decoder of the UNet. I wonder if this configuration was tried, and if yes, why it wasn't successful in this case?

---

> ### Author Response · Authors · 2023-11-22
> **Response to Reviewer dE3Q**
>
> We sincerely appreciate your valuable comments! Below is our response.
>
> ### **1. Novelty**
>
> We would like to respectfully address the concern raised about the novelty of our work. After carefully considering the comparison with the cited papers, we believe it may be *unfair* to characterize our work as lacking novelty. Our approach introduces a **seamless plug-and-play module** without modifying the diffusion backbone which distinguishes it from the referenced literature.
>
> It's important to note that the referenced papers all tackle *different* tasks from ours, such as person image synthesis [i,ii,vi], pose/motion-guided video transfer [iii,iv], exemplar-guided image generation [v], and consistent view synthesis [vii].
> Furthermore, some of these works **do not utilize diffusion models** [iv,v,vi], and one specifically **does not employ cross-attention** [iii].
> The others [i,ii,vii] **build and train raw UNet-based frameworks** with cross-attention blocks, resembling the text cross-attention blocks in LDMs and necessitating an encoder module before the cross-attention blocks.
> As recognized by the reviewer, our work stands out as **the first to explore image cross-attention in LDMs**. The distinctive aspect of our contribution resides in the design of the visual condition module, which can be seamlessly plugged into the text condition blocks. We have identified that cross-attention, a commonly used network design, is particularly well-suited for this module. Our plug-and-play method empowers us to **freeze** the pretrained diffusion model, focusing exclusively on training **the lightweight visual condition module**.
>
> ### **2. Training with more data**
>
> The described idea of training the module with a large dataset is actually similar to **encoder-based methods** [1,2,3]. We agree that training cross-attention layers on a large dataset may potentially bring improvement. It's crucial to note, however, that training such a module requires a significantly **larger amount of data within the same domain**, rather than relying on all the images in our paper.
> Additionally, encoder-based methods are often **limited to specific domains**; for instance, [1] trains on FFHQ, a dataset comprising 70k images specifically for the face domain.
> In contrast, our method is **lightweight**, trained on only a few images. A comparison between our method and an encoder-based model, E4T [1], is included in **Figure 23 in Appendix I**, illustrating the superior performance of our model.
> We would like to kindly inform the reviewer that, at present, *no* method, including encoder-based models, can perfectly reconstruct the exact appearance of the target object concept, especially when considering complex and nuanced details. We are committed to ongoing efforts for further improvement in this aspect.
>
> ### **3. Module incorporation in both the encoder and decoder**
>
> We tried this configuration of incorporating the module in both the encoder and decoder, which demonstrated performance **equivalent to** incorporating the module solely in the decoder. Therefore, we opt to apply the visual condition exclusively in the decoder to **achieve a reduction in parameters** and **enhance training efficiency**. We have included a comparison in **Figure 8 in Appendix A**, showcasing the incorporation of the visual condition module in both the encoder and decoder.
>
> [1] Gal, Rinon, et al. "Encoder-based domain tuning for fast personalization of text-to-image models." ACM Transactions on Graphics (TOG) 42.4 (2023): 1-13.
> [2] Shi, Jing, et al. "Instantbooth: Personalized text-to-image generation without test-time finetuning." arXiv preprint arXiv:2304.03411 (2023).
> [3] Jia, Xuhui, et al. "Taming encoder for zero fine-tuning image customization with text-to-image diffusion models." arXiv preprint arXiv:2304.02642 (2023).

---

### Official Review · Reviewer_cbrM · 2023-11-01

**Soundness:** 2 fair
**Presentation:** 2 fair
**Contribution:** 2 fair
**Rating:** 5
**Confidence:** 4

**Summary:**

This paper proposes a method for customizing text-to-image generation models, which requires less number of parameters to be tuned and less training time compared to related works.

**Strengths:**

The paper proposes to introduce extra attention modules, which can introduce new concept into the diffusion process. The introduced attention modules contain much less parameters compared to the whole diffusion model, leading to more efficient fine-tuning.

Only fine-tuning introduced attention modules have the advantage of maintaining the original capability of pre-trained models, which might be important.

According to the experiment results shown in the paper, better results are obtained compared to vanilla DreamBooth. The needed training time is also much less.

**Weaknesses:**

The major concern is on the experiments, why do the authors only use 20 unique concepts from the Textual Inversion, DreamBooth, Custom Diffusion, rather than use a union of their testing samples or directly use the DreamBench dataset proposed in DreamBooth paper?

One important related work is missing [1], which requires fine-tuning less number of parameters compared to LoRA, and maintains the original capability of the pre-trained model. As shown in the paper, the method is also very stable (please see question in next section).

The low $T_{CLIP}$ score may indicate unsatisfactory edit-ability, thus more qualitative results in terms of complicated style change are suggested.

Although encoder-based methods are not directly related to the proposed method, comparison and discussion are strongly suggested. Especially considering the fact that encoder-based methods normally require much less fine-tuning time or are even tuning-free (although they need pre-training).

[1]. Controlling Text-to-Image Diffusion by Orthogonal Finetuning. Zeju Qiu, Weiyang Liu, Haiwen Feng, Yuxuan Xue, Yao Feng, Zhen Liu, Dan Zhang, Adrian Weller, Bernhard Schölkopf.

**Questions:**

Have the author considered using ground-truth mask in computing the diffusion loss (with a pre-trained model like SAM[1]), or forcing the attention map to be aligned with the mask?

In DreamBooth, when the model is fine-tuned for too much iterations, the model performance may degenerate even when augmentation data and prior loss is used. Will this also happen with the proposed method? Specifically, generated results with respect to different iterations are suggested to be shown, especially when the number of training steps are very large. This result is important as related work OFT is shown to be stable even after thousands of fine-tuning steps.

[1]. Segment Anything. Alexander Kirillov, Eric Mintun, Nikhila Ravi, Hanzi Mao, Chloe Rolland, Laura Gustafson, Tete Xiao, Spencer Whitehead, Alexander C. Berg, Wan-Yen Lo, Piotr Dollár, Ross Girshick.

---

> ### Author Response · Authors · 2023-11-22
> **Response to Reviewer cbrM**
>
> Thank you for your constructive comments! Below is our response.
>
> ***W1: Evaluation dataset***
>
> We select an approximately *middle number* of concepts from each method: 9 released by Textual Inversion, 30 released by DreamBooth, and 7 released by Custom Diffusion, for a fair comparison.
> Following the suggestion, we evaluate all methods on **30 concepts** from the DreamBooth dataset:
> |                   |   $I_{DINO}$   |   $I_{CLIP}$  |   $T_{CLIP}$   |
> |-------------------|:----------:|:---------:|:----------:|
> | DreamBooth        |  **0.6470**  |  0.8064 | 0.2384 |
> | Custom Diffusion  |  0.5956 |  0.7755 |  **0.2485** |
> | Textual Inversion | 0.5417 |  0.7677  |   0.2140  |
> | Ours              |   0.6406  | **0.8080** |   0.2268  |
>
> The results are **similar** to the testing conducted on 20 mixed concepts, as reported in the paper. ViCo achieves image similarity comparable to DreamBooth and significantly outperforms Textual Inversion in all evaluation metrics.
>
> ***W2&Q2: Comparison and discussion on OFT [1] & Large number of training steps***
>
> Thanks for pointing this concurrent work to us. We have included **the discussion on OFT [1]** in the related work (**Section 2**).
>
> OFT and ViCo are designed for different purposes. OFT is a fine-tuning method and can be used for multiple downstream tasks, including DreamBooth, while ViCo is a plug-and-play method that can be applied to any pretrained diffusion backbone. Therefore, OFT and ViCo are not direct competitors and they can be seamlessly used *together*.
>
> Following the suggestion,  we have added experimental results comparing **ViCo**, **OFT**, and **ViCo with OFT** with a **large number of training steps**. Please refer to **Figure 22 in Appendix I** for more detailed information. We implemented the constrained OFT using the official code. In cases where the number of training steps is very large, ViCo may *not* exhibit image distortion but could potentially overfit to the training image. However, this issue can be effectively addressed by combining ViCo with OFT, resulting in the best overall results.
>
> ***W3: More qualitative results of complicated style change***
>
> Thanks for the suggestion. We have added more qualitative results in terms of complicated style change in **Figure 28 in Appendix J**.
>
> In our qualitative results, we **do not observe** unsatisfactory editability. The slightly lower $T_{CLIP}$ score can be attributed to the fact that the text information may appear less distinctive compared to DB and CD, rather than being unable to appear at all. This is mainly due to the dominance of the single token "S*" in the textual embedding space, which differs from the "[V] [category]" format used by DB and CD.
>
> ***W4: Comparison and discussion on the encoder-based method***
>
> We highlighted the difference between encoder-based models and our method in the related work (**Section 2**).
>
> We have followed the suggestion to provide a more in-depth comparison and discussion, focusing on a representative encoder-based method, Encoder for Tuning (**E4T**) [2]. In **Figure 23 in Appendix I**, we have added a comparison between our model and E4T [2]. We implemented E4T using the unofficial code (as no official code is available) and employed their provided model that is pretrained on large-scale face images. Note that the E4T involves a 2-stage training process: initial pretraining on a large-scale dataset of a specific category domain, such as human faces, followed by fine-tuning on a single concept image. Our results reveal that ViCo surpasses E4T in its ability to effectively preserve facial details.
>
> ***Q1: Ground-truth masks and Attention alignment***
>
> Yes, we have considered it. We tried using the **groud-truth mask** (generated by SAM) for masking. We have also considered **attention alignment** that forces the S* cross-attention map to align with the mask using MSE. Please kindly refer to **Figure 9 in Appendix A** for the comparison results.
> We compared 4 model settings:
> 1. Masking with SAM masks
> 2. Alignment with SAM masks
> 3. Alignment with automatic masks
> 4. Masking with automatic masks (**our approach**)
>
> Our results indicate that our automatic mask performs **comparably to** the ground-truth mask, highlighting its effectiveness. Moreover, our masking strategy exhibits a significant **performance advantage** over attention alignment.
>
> [1] Qiu, Zeju, et al. "Controlling text-to-image diffusion by orthogonal finetuning." In NeurIPS, 2023.
> [2] Gal, Rinon, et al. "Encoder-based domain tuning for fast personalization of text-to-image models." ACM Transactions on Graphics (TOG) 42.4 (2023): 1-13.

---

### Official Review · Reviewer_xPT1 · 2023-11-02

**Soundness:** 3 good
**Presentation:** 3 good
**Contribution:** 3 good
**Rating:** 6
**Confidence:** 2

**Summary:**

This paper proposes a text2image personalization method that learns the personalization text embedding and the proposed image attention. "Image attention" is a cross-attention module to integrate visual conditions into the denoising process for capturing object-specific semantics. A mask that is derived from the cross-attention map between reference image and text is applied to the "image attention" used to focus more on the object of the reference image.

**Strengths:**

- The method only requires learning relatively few parameters to effectively incorporate the information from reference image for personalized text2image generation.
- The results are very favorable compared to existing methods like DreamBooth and Textual Inversion, while having a low training time cost.
- Good ablation study and analysis provided in the paper.
- The paper is quite transparent and information-rich in many ways, which is good for reproducibility purposes.

**Weaknesses:**

- In Table 4, the improvement introduced by masking is not so significant.
- The method is incapable of using multiple reference images during inference, for more robust generation.
- Apparently, the method only works with images that have a single reference primary object.

**Questions:**

1. Do you use the same reference image for different model variations in the evaluation (especially T4)?

---

> ### Author Response · Authors · 2023-11-22
> **Response to Reviewer xPT1**
>
> We are grateful for your valuable feedback! Below is our response.
>
> >W1: In Table 4, the improvement introduced by masking is not so significant.
>
> We would like to bring to the attention of the reviewers that the quantitative comparison might not clearly demonstrate the object refinement achieved by the masking strategy, as it refines more visual details beyond just the texture characteristics of the object. These additional refinements are challenging to distinguish significantly through evaluation by visual encoders.
>
> However, it is important to note that our proposed image cross-attention already possesses object capture capabilities during training, exhibiting high quantitative performance. The masking strategy additionally provides an explicit method to filter out the background of the reference image seamlessly in our framework. This rectifies certain cases where the generation without the mask may be unsatisfactory. For specific examples, we encourage the reviewer to refer to **Figure 5 (b)**. In cases where a mask is absent, the generation may solely capture the shape and texture of the object, neglecting important shape constraints. Introducing an automatic mask overcomes this issue and further enhances the object's identity in the generated images.
>
> >W2: The method is incapable of using multiple reference images during inference, for more robust generation.
>
> We thank the reviewer for bringing up the point of using multiple reference images during inference. We would like to clarify that our framework indeed **supports an arbitrary number of reference images (one or multiple)** without any change to the architecture.
> This capability is achieved by **concatenating tokens of multiple images** and passing them through the existing image cross-attention.
> The flexibility of our framework lies in the fact that the number of input tokens in the image cross-attention can be *variable*, accommodating any number of tokens from the concatenated reference images. This attribute allows us to **flexibly use an arbitrary number of reference images**.
>
> We have conducted a comparison between using two reference images and using a single reference image for generation, as shown in **Figure 12 in Appendix C**. Our findings demonstrate that the results obtained using two reference images are similar to those achieved when using a single reference image. This observation indicates the *robustness* of our visual condition module to variations in the reference image. We appreciate the reviewer's recognition of this important characteristic of our framework.
>
> >W3: Apparently, the method only works with images that have a single reference primary object.
>
> This is a **common limitation** for the majority of existing works, e.g., Textual Inversion, DreamBooth, and Custom Diffusion. Current research on personalized text-to-image generation mainly focuses on a single object. Learning from an image that has multiple objects is still an open problem for the community.
> Currently, the only work exploring this challenging problem is [1] which heavily relies on predefined object masks. Our method can also be adapted to this problem by manually providing object masks as done in [1]. We are indeed studying a more generalized case, in which no object masks are given, in our current research.
>
> >Q1: Do you use the same reference image for different model variations in the evaluation (especially T4)?
>
> Yes. For all experiments in the paper, we use **the same reference image** for each dataset.
> We have further clarified this in the revised paper.
>
> [1] Avrahami, Omri, et al. "Break-A-Scene: Extracting Multiple Concepts from a Single Image." In SIGGRAPH Asia, 2023.

---

### Author Response · Authors · 2023-11-22
**Global comment**

Dear reviewers,

**Thank you for your constructive comments!**

We appreciate the reviewers for recognizing the merits of our paper: good ablation study and transparent writing (**xPT1**), a lighter-weight and faster module (**xPT1**, **cbrM**, **dE3Q**), favorable performance (**xPT1**, **cbrM**, **dE3Q**), maintaining diffusion model capabilities (**cbrM**), an efficient non-parametric mask mechanism (**Z7sX**), and a flexible and easy-to-deploy framework (**Z7sX**) without the need for heavy preprocessing or annotations (**Z7sX**).

We have carefully addressed individual concerns in the response to each reviewer, incorporating the suggestions to enhance the paper. The revised version includes:

1. A comparison between using one or two reference images (**Figure 12 in Appendix C**).
2. A clarification regarding the use of the same reference image for each dataset (**Section 3.4**).
3. A discussion on OFT [1] in the related work (**Section 2**).
4. A comparison among ViCo, OFT, and ViCo with OFT (**Figure 22 in Appendix I**).
5. Additional qualitative results showcasing complicated style changes (**Figure 28 in Appendix J**).
6. A comparison with E4T [2] (**Figure 23 in Appendix I**).
7. A comparison involving ground-truth masks and a distinction between masking strategy and attention alignment (**Figure 9 in Appendix A**).
8. A comparison with module incorporation in both the encoder and decoder layers (**Figure 8 in Appendix A**).

The revisions are highlighted in blue, further strengthening the paper based on the constructive comments.

**If you still have any further concerns, we would be happy to discuss them!**

[1] Qiu, Zeju, et al. "Controlling text-to-image diffusion by orthogonal finetuning." In NeurIPS, 2023.
[2] Gal, Rinon, et al. "Encoder-based domain tuning for fast personalization of text-to-image models." ACM Transactions on Graphics (TOG) 42.4 (2023): 1-13.

---

### Meta-Review · Area_Chair_CYcY · 2023-12-06

**Metareview:**

This paper presents ViCo, a plug-and-play method that integrates visual conditions into personalized text-to-image generation. Four reviewers provided ratings of 6, 5, 5, 6.  Positive points include the efficient approach and favorable performance compared to baselines. Some common concerns include limited novelty, missing baselines, and issues with some experiments/datasets. The rebuttal partly addressed the concerns regarding missing baselines and experiments/datasets. However, the limited novelty remained a concern for some reviewers, including Z7sX who, based on the other reviews and authors' rebuttal, also made this point. Overall, this is a borderline paper with split reviews and none of the reviewers strongly advocating acceptance. The paper, rebuttal, discussion, and author messages were carefully discussed among the ACs, and the ACs agree that the paper does not yet meet the bar for ICLR acceptance, mainly due to limited technical contribution. The ACs would like to encourage the authors to improve the paper and resubmit to another conference.

**Justification For Why Not Higher Score:**

Based on the weaknesses mentioned above, the paper does not meet the bar for acceptance.

**Justification For Why Not Lower Score:**

N/A

---

### Decision · Program_Chairs · 2024-01-16

Reject